



# Introducing CRYOWRF v1.0: Multiscale atmospheric flow simulations with advanced snow cover modelling

Varun Sharma[1,2,3], Franziska Gerber[1,2], and Michael Lehning[1,2]

[1]School of Architecture, Civil and Environmental Engineering, Swiss Federal Institute of Technology, Lausanne, Switzerland
[2]WSL Institute for Snow and Avalanche Research SLF, Davos, Switzerland
[3]Sunwell Sàrl, Lausanne, Switzerland

**Correspondence:** Varun Sharma (varun.sharma@epfl.ch)

**Abstract.** Accurately simulating snow-cover dynamics and the snow-atmosphere coupling is of major importance for topics as wide-ranging as water resources, natural hazards and climate change impacts with consequences for sea-level rise. We present a new modelling framework for atmospheric flow simulations for cryospheric regions called CRYOWRF. CRYOWRF couples the state-of-the-art and widely used atmospheric model WRF, with the detailed snow-cover model SNOWPACK. CRYOWRF makes it feasible to simulate dynamics of a large number of snow layers governed by grain-scale prognostic variables with *online* coupling to the atmosphere for multiscale simulations from the synoptic to the turbulent scales. Additionally, a new blowing snow scheme is introduced in CRYOWRF and is discussed in detail. CRYOWRF's technical design goals and model capabilities are described and performance costs are shown to compare favourably with existing land surface schemes. Three case studies showcasing envisaged use-cases for CRYOWRF for polar ice sheets and alpine snowpacks are provided to equip potential users with templates for their research. Finally, the future road-map for CRYOWRF's development and usage is discussed.

## 1 Introduction

The cryosphere consists of regions of the Earth where snow and ice cover the surface for some reasonable period during the course of the year. It consists of regions with seasonal snow cover, frozen lakes and rivers, glaciers, continental ice sheets, sea ice and permafrost, and covers a mean total area of $70 \times 10^6$ km$^2$ (14% of Earth's surface area) with sub-annual fluctuations between $82 \times 10^6$ km$^2$ and $57 \times 10^6$ km$^2$ depending on the Northern Hemisphere winter season. Incidentally, the seasonal snow-cover, mostly in the Northern Hemisphere, represents the largest surface area (and shortest timescale) of any component of the cryosphere. On the other hand, perennial components of the snow cover, namely glaciers and polar ice sheets in Greenland and Antarctica cover approximately $16 \times 10^6$ km$^2$ of surface area. Most important is the fact that 80 % of the freshwater on Earth is held in these components with Antarctica being the largest store, with 90% of the freshwater ice mass (Ohmura, 2004; Hock et al., 2009).



From the perspective of land-atmosphere interaction, there is a stark contrast between snow-free and snow-covered regions in terms of mass and energy fluxes. These differences emerge due to the defining material properties of snow: high albedo, low thermal conductivity, large latent heat for phase change and low mechanical roughness (Arduini et al., 2019). Betts et al. (2014) for example showed that simply changing between snow-covered and snow-free conditions results in a 10 K difference in 2-m temperature climatology with all other forcings kept constant. There is thus, a vast area of the planet where accurately modelling the dynamics of snow-cover and its special interaction with the atmosphere is of primary importance.

The cryosphere is studied at different scales with different motivations. At the largest, planetary scale, accounting for the cryosphere is critical since it modulates the energy and water budget of the Earth System (Vaughan et al., 2003; Frei et al., 2012). At a regional (or *catchment* scale), snow cover dynamics play a significant role for water resources for agriculture (Barnett et al., 2005) and hydroelectricity (Schaefli, 2015; Dujardin et al., 2021), on account of its delayed release of water (melting phase) following a prolonged winter season consisting of the accumulation and storage phases. The melting phase of the snow cover is particularly strongly correlated with hazards such as spring floods (Bloeschl et al., 2017) and landslides (Harpold and Kohler, 2017). In some countries with a well developed ski industry, snow modelling is used extensively to guide the planning and management of skiing assets and more importantly, for predicting avalanches (Hanzer et al., 2020).

Various numerical models with varying degrees of complexity have been developed to simulate the dynamics of snow cover. Snow models are a primary component of climate models, which until recently had a relatively simplistic description of snow. For example, most participating models in CMIP5 (Taylor et al., 2012) used a *single layer* approach, where snow is considered as an additional layer covering the soil. In these models snow properties such as albedo evolve via empirical parametrizations. Models following this approach are termed as *low complexity* models and continue to be used in many numerical weather prediction (NWP) models (for example, the COSMO model's TERRA land surface model, Doms et al., 2011, used in many countries operationally in Europe). As explained by Jin et al. (1999) and Arduini et al. (2019), such models are able to represent only a single time-scale of snow-cover dynamics and are prone to errors at all timescales, ranging from the sub-diurnal to the seasonal. This is due to neglecting the strong vertical gradients of density and temperature typically found in snowpacks not to speak about localized phase changes. Single layer models have been shown to inaccurately simulate snow depth and mass balance as well as allied issues like soil freezing, onset of melt etc. (Saha et al., 2017; Dutra et al., 2012). While use of data assimilation can ameliorate these issues for short term NWP applications, sub-seasonal, seasonal and climate scale forecasts are severely affected by such issues.

At the other end of the complexity spectrum, sophisticated snow models used for avalanche prediction were developed (Krinner et al., 2018). These models have a *multi-layer* description of snow with prognostic description of grain-scale characteristics of snow such as grain diameter, sphericity and dendricity and inter-grain bond sizes and coordination numbers. Various bulk material properties such as hydraulic and thermal conductivity, albedo, compaction and settling rates (through viscosity parametrizations) are determined directly from grain-scale properties. Such level of detail is required for avalanche prediction since a single, thin layer of low-density snow, with particular grain characteristics can mechanically destabilize the entire snowpack. Such snow models can be denoted as *high complexity* models. Examples of such models are the CROCUS model from France (Vionnet et al., 2012) and SNOWPACK (more on this model in the following paragraphs).



There are models termed as *intermediate* complexity models that still have the multi-layer representation of snow but rely on parametrizations to derive the bulk properties of snow layers without requiring grain scale information. These models therefore provide the possibility of representing vertical stratification within snowpacks, while avoiding the relatively expensive computational costs of high complexity models. These models have been shown to significantly improve the representation of the snow-cover and its impact on soil as well as the atmosphere in various studies (Xue et al., 2003; Burke et al., 2013; Decharme et al., 2016). It is not surprising therefore that just between the CMIP5 and its successor, the CMIP6 (Eyring et al., 2016), most participating models have adopted multi-layer, intermediate complexity models (see also, the related, Earth System Model Snow Intercomparison Project, ESM-SnowMIP; Krinner et al., 2018; Menard et al., 2019). Only recently, the European Center of Medium-range Weather Forecasts (ECMWF) Integrated Forecasting System (IFS) has transitioned from a single-layer to a multi-layer treatment of snow (Arduini et al., 2019), heralding the adoption of intermediate complexity snow models within operational NWP.

In spite of the progression in the complexity of snow treatment in NWP and climate models, the continual increase in computational power as well as resources, along with the scope and need for further improvement, leads us to believe that coupling between NWP or climate models and high complexity, advanced snow models is both feasible and required, in particular for resolving complex surface processes such as drifting and blowing snow. In this work, we describe the first version of CRYOWRF, a fully coupled atmosphere-snowpack solver suitable for simulations in both alpine as well as polar regions. CRYOWRF brings together two state-of-the art models, with WRF (the Weather Research and Forecasting model) being the atmospheric core of the model while SNOWPACK, a *high complexity* snow model, acts as the land-surface model (LSM).

This development brings together compelling advantages: WRF is a widely used community model with multi-scale capabilities on account of its non-hydrostatic dynamical core along with domain nesting. This allows simulations from global scale (with grids sizes of 50-100 km) down to turbulence scales (with grid sizes of 10-50 m) using the same software package. WRF is continuously updated with newer, improved physical parameterizations for all the different model components such as radiation, microphysics, boundary layer schemes etc. From a users' perspective, there are many tutorials available online to learn the entire workflow from compilation, pre-processing of input data and running the model to post-processing model outputs. Thus any development extending WRF is likely to reach a lot of potential users.

SNOWPACK is a well-established *high complexity* snow model developed at the WSL Institute for Snow and Avalanche Research (SLF), Davos, Switzerland. It was originally developed for avalanche forecasting and is in fact the operational model for this purpose in Switzerland. Since its original development nearly two decades ago, SNOWPACK has been continuously updated to include new physics and has been applied in different topics of cryospheric research such as snow hydrology (Wever et al., 2015, 2016a, b; Brauchli et al., 2017), sea-ice thermodynamics (Wever et al., 2020), snow-forest interaction (Gouttevin et al., 2015), ice-sheets mass balance and thermodynamics (Steger et al., 2017b, a; van Wessem et al., 2021; Keenan et al., 2021), and climate-change induced impact assessments on snow and snow hydrology (Bavay et al., 2013, 2009; Marty et al., 2017). In all the above applications, SNOWPACK has been forced either with meteorological measurement data





or NWP/climate model outputs in an *offline* setting. CRYOWRF is the first implementation of an *online* coupling between SNOWPACK and an atmospheric model.

A significant direct physical coupling between the snowpack and the atmosphere is through drifting and blowing snow and
its effect on the mass and energy balance of the near surface atmosphere. Drifting and blowing snow redistributes surface snow spatially, while inducing enhanced sublimation of (and sometimes deposition on) snow grains and thereby saturating (drying) the near surface air. Blowing snow is estimated to affect nearly all cryospheric environments (Filhol and Sturm, 2015) and there is evidence of their relevance for topics as disparate as avalanche prediction (Lehning and Fierz, 2008), road safety (Tabler, 2003), hydrology (Groot Zwaaftink et al., 2014), interpretation of ice cores (Birnbaum et al., 2010), remote sensing
of snow-covered areas (Warren et al., 1998) and thermo-mechanical dynamics of sea-ice packs (Petrich et al., 2012). In spite of its ubiquitous importance in cryospheric regions, blowing snow models are not included in any of the participating models in CMIP6. There are implementations of blowing snow models in regional scale climate models (RCMs) such as RACMO (van Wessem et al., 2018; Lenaerts et al., 2012) and MAR (Amory et al., 2015, for polar regions) and Meso-NH (Vionnet et al., 2012, for alpine regions). Publicly available versions of WRF do not include any model of this important phenomenon.
Thus, apart from the coupling between WRF and SNOWPACK, a completely novel blowing snow scheme is implemented and released publicly as a part of CRYOWRF.

In Sec. 2 we describe the three models underlying CRYOWRF, namely WRF, SNOWPACK and in particular, the blowing snow scheme, in more detail. We discuss the model capabilities of CRYOWRF as well as its computational costs in Sec. 3. Three case studies are then described that exemplify the range of applications for CRYOWRF: a continent-scale simulation over
Antarctica for mass balance studies (Sec. 4.1), a multi-scale, three domain simulation (1 km resolution of the inner-most nest) of a hydraulic jump in the vicinity of the Dumont d'Urville station on the East Antarctica coast (Sec. 4.2) and a valley-scale simulation in the Swiss Alps surrounding Davos (Sec. 4.3) for avalanche prediction and hydrology applications. We conclude this article with a summary and outlook for future developments of CRYOWRF (Sec. 5).

## 2 Models and coupling strategy

### 2.1 Weather Research and Forecasting Model (WRF)

The Weather Research and Forecasting Model (WRF) is an advanced atmospheric modeling system aimed at both operational numerical weather prediction as well as research activities. It was originally developed and continues to be managed by the the NCAR/MMM (Mesocale and Microscale Meteorology division, National Center for Atmospheric Research) from the USA. Due to its completely open-source nature and a large user community, it has significant contributions from research groups and
weather services from around the world.

The *Advanced Research WRF* (ARW) version of the model (Skamarock et al., 2019) solves the fully-compressible, Eulerian non-hydrostatic equations for the three velocity components in Cartesian coordinates in the horizontal along with a terrain-following, hybrid sigma-pressure vertical coordinate system with a constant pressure surface as the top boundary. Additional prognostic variables are the moist potential temperature and the geopotential. Optionally, depending on the microphysics and





chemistry schemes chosen, dynamics of different hydrometeor and chemical species are also prognostically solved for, by
using advection-diffusion equations with additional source/sink terms. The primary numerical method is finite-differencing for
spatial derivatives on an Arakawa-C grid with a flexible grid stretching in the vertical. Time-stepping is done using second
or third order Runge-Kutta schemes with faster, internal time-stepping to solve for the acoustic modes. The model is fully
parallelized with a hybrid MPI-OPENMP implementation for both the solver as well as the input-output (I/O) processes.

Additional important technical capabilities are ease of performing multi-scale atmospheric simulations with both one-way and
two-ways nesting, and start-stop-restart capabilities for all nests. The multiscale capabilities and its community driven nature
made WRF the ideal choice as the atmospheric core of CRYOWRF. CRYOWRF v1.0 is derived from the ARW-WRF v4.2.1
with plans to continuously update to newer versions of WRF as and when they are released.

## 2.2  SNOWPACK

SNOWPACK, originally developed for avalanche warning (Lehning et al., 1999), has evolved into a multi-purpose model for
cryospheric snow-atmosphere interactions with modules for soil/permafrost (Haberkorn et al., 2017), vegetation (Gouttevin
et al., 2015) and sea ice (Wever et al., 2020). The core SNOWPACK module solves the heat equation on a dynamic finite ele-
ment mesh, which evolves with mass changes of ice and snow such that the identity of layers is preserved. Using transient snow
microstructure to determine snow properties such as viscosity, thermal conductivity or albedo, a very detailed representation of

snow processes from settling to phase changes and water transport (Wever et al., 2014) results. Upper boundary conditions for
mass and heat include many options e.g. for initial snow density (Zwaaftink et al., 2013), for snow erosion (Lehning and Fierz,
2008), or for stability corrections (Schlogl et al., 2017). The one-dimensional SNOWPACK model has previously been coupled
(in an *offline* sense) with a three-dimensional solver in the atmosphere specifically for snow transport and a spatial description
of snow - atmosphere interaction (Lehning et al., 2008). The coupling to WRF is similar in concept as described further, with

the main difference being transitioning from an offline to an online approach. Grain-scale information of the snowpack is
critical for modelling drifting and blowing snow that is strongly governed by the grain size and the cohesion between grains
(Comola and Lehning, 2017). This is one of the main motivations for coupling WRF and SNOWPACK. Additionally, online
coupling allows for direct exploration of various feedback processes that the offline approach cannot describe.

## 2.3  Blowing snow Model (BSM)

When wind speeds are larger than a certain threshold value, the shear stress at the snow surface is sufficiently high to dislodge
snow grains and entrain them into the air. The larger (and heavier) particles are transported along the surface in ballistic
trajectories, rebounding over the snow surface and dislodging additional grains due to collision. This phenomenon, known as
*saltation* occurs in a very shallow layer with a thickness of $\mathcal{O}(10cm)$. The smaller particles are instead *picked up* by turbulent
eddies that transport the particles to much greater heights and over much larger horizontal distances, referred to as *suspension*.

Suspension is quite similar in its dynamics to dust or cloud ice. Saltation and suspension together is referred to as *blowing
snow* in this study.





In large scale atmospheric models, it is impossible to have vertical resolutions close to the surface that can resolve the saltation layer and thus, it is only the suspension layer that is explicitly solved for with the saltation layer being completely parameterized and acting as a lower boundary condition for the suspension layer. Thus, in what follows, the *blowing snow model*, actually refers to the model for snow suspension.

The blowing snow model implemented in CRYOWRF is a double-moment scheme that solves prognostic equations for the mass $\left(q_{bs} \left[\mathrm{kg\,kg^{-1}}\right]\right)$ and number $\left(N_{bs} \left[\mathrm{kg^{-1}}\right]\right)$ mixing ratios of blowing snow particles. These equations are essentially Eulerian advection-diffusion type equations along with phase changes of sublimation and vapor deposition. They are quite similar to those used to solve for solid-phase hydrometeors such as cloud ice or snow. The blowing snow model is described in much greater detail as compared to the previous sections as it is a novel parametrization complementing WRF and SNOWPACK, which individually are established and well known stand-alone modelling systems.

The blowing snow scheme in CRYOWRF closely follows the framework by Dery and Yau (2002) and the implementation of blowing snow dynamics in Meso-NH (Lafore et al., 1998; Lac et al., 2018) by Vionnet et al. (2014). The principle difference between our implementation and these previous efforts are in the details of calculating terminal fall velocities and threshold friction velocity for snow transport. These changes are described in the following paragraphs. The intended goal of this section is to describe the implementation of the blowing snow scheme for potential users of CRYOWRF and not to discuss in detail, the scientific or physical basis of the choices made and the model's comparison with other previously published models. Those discussions are kept for a future publication.

The equations in Einstein notation for $q_{bs}$ and $N_{bs}$ respectively are

$$\frac{\partial q_{bs}}{\partial t} + u_i \frac{\partial q_{bs}}{\partial x_i} = -K_{bs} \frac{\partial^2 q_{bs}}{\partial x_i^2} \delta_{i3} + \frac{\partial}{\partial x_i} \left(q_{bs} V_q \delta_{i3}\right) + S_q \,, \tag{1a}$$

$$\frac{\partial N_{bs}}{\partial t} + u_i \frac{\partial N_{bs}}{\partial x_i} = -K_{bs} \frac{\partial^2 N_{bs}}{\partial x_i^2} \delta_{i3} + \frac{\partial}{\partial x_i} \left(N_{bs} V_N \delta_{i3}\right) + S_N \,, \tag{1b}$$

where $u_{i=1,2,3} = (u, v, w)$ are the zonal, meridional and vertical velocity components of the air respectively. $K_{bs}$ is the turbulent diffusion coefficient for blowing snow particles, $V_q$ and $V_N$ are the mass and number-weighted terminal fall velocities of blowing snow particles and the $S_q$ and $S_N$ are sink (source) terms to account for sublimation (deposition) of blowing snow particles.

There is a lack of clarity about the values of $K_{bs}$. In CRYOWRF, $K_{bs}$ is taken to be the same as that for turbulent diffusion coefficient for heat following Dery and Yau (1999, 2002). In Vionnet et al. (2014), it is modified to 4 times the equivalent coefficient for heat without explicit reasoning for the modification. Further research, perhaps using dedicated large-eddy simulations of the sort in Sharma et al. (2018) can be used to derive parametrizations for this parameter.

Linking the mass and number mixing ratios is the two-parameter gamma distribution, similar to that used in various double-moment microphysics schemes (Morrison and Grabowski, 2008; Thompson et al., 2008) and indeed in different blowing snow models (Dery and Yau, 2002; Vionnet et al., 2014). The snow particle-size distribution is given by

$$n\left(D\right) = N_{bs} \,\rho_{air} \frac{\lambda^\alpha D^{\alpha-1} e^{-D\lambda}}{\Gamma\left(\alpha\right)} \,, \tag{2}$$





where $D$ is the particle diameter, $\Gamma$ is the Gamma function, $\alpha$ and $\lambda$ are the shape and inverse-scale parameters, respectively, and $\rho_{air}$ is the air density. The above distribution implies the assumption that blowing snow particles are spherical. The $\alpha$ parameter is usually assumed to be between 2 and 3 and in CRYOWRF, it is chosen to be equal to 3, following Vionnet et al. (2014). Given the assumption of spherical particles and a prescribed $\alpha$, the parameters $\lambda$ and the mean particle diameter $\overline{D}$ can be diagnosed from the mixing ratios as,

$$\lambda = \left[ \frac{\pi \rho_i}{6} \frac{N_{bs}}{q_{bs}} \frac{\Gamma(\alpha+3)}{\Gamma(\alpha)} \right]^{1/3}, \quad \overline{D} = \frac{\alpha}{\lambda}, \tag{3}$$

where $\rho_i$ is the density of individual blowing snow particles (taken to be that of ice). With $\alpha$ known (or assumed) and $\lambda$ derived, the full distribution is available at every node of the numerical grid. Note that $\alpha$ is assumed to not vary with height following Dery and Yau (2001) and Vionnet et al. (2014). This is only partially supported by (limited) measurements (Mann et al., 2000; Nishimura and Nemoto, 2005) but more field-data is required to conclude about this assumption.

For the calculation of terminal fall velocities, we deviate from the implementations of Dery and Yau (2002) and Vionnet et al. (2014) and instead follow the Best Number ($X$) based derivations of Mitchell (1996), Khvorostyanov and Curry (2002) and Sulia and Harrington (2011). Incidentally, this approach is also used in the latest model for cold clouds, ISHMAEL (Jensen et al., 2017). The notable feature of ISHMAEL is that its formulation for fall velocities is applicable for all shapes of the ice habit with aspect ratios ranging from planer to spherical and columnar ice particles. Considering our assumption of spherical blowing snow particles, the mathematical expressions for our purposes are significantly simplified as described below.

The Best Number ($X$) is defined as,

$$X = \frac{4}{3} \frac{D^3 \rho_{air} (\rho_i - \rho_{air}) g}{\eta^2}, \tag{4}$$

where, $D$ is the particle diameter, $\eta$ is the dynamic viscosity of air, $\rho_{air}$ and $\rho_i$ are the density of air and blowing snow particles, respectively, and $g$ is the gravitational acceleration. The Best Number is related to the Reynolds number of the particles following Mitchell (1996) as,

$$Re = a_m X^{b_m}. \tag{5}$$

Here, $a_m$ and $b_m$ are fitting coefficients derived by Mitchell (1996) and functions only of the Best Number.

The first step to calculate the mass and number weighted terminal fall velocities is to derive an expression for terminal fall velocity as a function of particle diameter. To do so, we begin by calculating the mean Best Number as,

$$\overline{X} = \frac{\int X(D) n(D) dD}{\int n(D) dD}, \tag{6a}$$

$$= \phi \frac{\Gamma(\alpha+3)}{\Gamma(\alpha)} \frac{1}{\lambda^3}, \text{where } \phi = \frac{4}{3} \frac{\rho_{air} (\rho_{ice} - \rho_{air}) g}{\eta^2} \tag{6b}$$

Next, the coefficients $a_m$ and $b_m$ are computed using the mean Best Number $\left( \overline{X} \right)$. In what follows, $a_m$ and $b_m$ are always understood to be $a_m \left( \overline{X} \right)$ and $b_m \left( \overline{X} \right)$, respectively. Using Eq. (5) and equating it to the standard form of the Reynolds' number,




the terminal fall velocity $V_t$ as a function of the particle diameter can be derived as,

$$\frac{\rho_{air} V_t D}{\eta} = a_m X^{b_m}, \tag{7a}$$

$$V_t = \phi^{b_m} \frac{a_m \eta}{\rho_{air}} D^{3b_m - 1}. \tag{7b}$$

From a technical perspective, Eq. (7b) is quite useful as it is a continuous and easily integrable function of $D$ of the form $(\cdot) D^{(\cdot)}$. From Eq. (7b), equations for $V_q$ and $V_N$ are derived as,

$$Vq = \frac{\int V_t(D) m(D) dD}{\int m(D) dD}, \tag{8}$$

$$= \Pi \frac{\Gamma(3b_m + 2 + \alpha)}{\Gamma(\alpha)\Gamma(\alpha + 3)} \lambda^{1 - 3b_m}, \text{where } \Pi = \frac{a_m \eta}{\rho_{air}} \phi^{b_m}, \tag{9}$$

and,

$$V_N = \frac{\int V_t(D) n(D) dD}{\int n(D) dD}, \tag{10a}$$

$$= \Pi \frac{\Gamma(3b_m - 1 + \alpha)}{\Gamma(\alpha)} \lambda^{1 - 3b_m}. \tag{10b}$$

In the above expressions, the particle mass distribution $m(D)$ is derivable from the particle size distribution $n(D)$ using the assumption of spherical particles.

Calculating the phase-change terms $S_q$ and $S_N$ in Eqs. (1) also begins by first describing the phase change of a solitary ice sphere as a function of its diameter is an easily integrable form and then integrating it over the number distribution $n(D)$. We use the well established Thorpe and Mason (1966) approach to calculate the sublimation (or deposition ) of an ice sphere as

$$S(D) = A D \sigma f_v, \tag{11}$$

where,

$$A = \frac{2\pi}{\frac{L_s}{K_{air} T_{air}} \left[\frac{L_s}{R_v T_{air}} - 1\right] + \frac{R_v T_{air}}{D_v e_{si}}}, \text{and} \tag{12a}$$

$$f_v = 0.78 + 0.308 Sc^{1/3} Re^{1/2}. \tag{12b}$$

In the above equations, $S(D)$ is the mass loss from a sphere of ice of diameter $(D)$ per unit time, $L_s$ is the latent heat of sublimation, $K_{air}$ is the molecular thermal conductivity of air, $T_{air}$ the air temperature, $R_v$ is the gas constant for water vapor, $D_v$ is the molecular diffusivity of water vapor and $e_{si}$ is the vapor pressure of the surrounding air with respect to ice. The effect of turbulence and its enhancement of the mass and energy transfer is encoded in the transfer coefficient $f_v$, which is related to the Schmidt number $(Sc)$ and the particle Reynolds' number $(Re)$. The formulation for $f_v$ is taken from Sharma et al. (2018) and similar formulations are used in Jensen et al. (2017) and Jafari et al. (2020). Finally the mass transfer is linked to relative humidity of the environment by the quantity $\sigma = [RH/100 - 1]$. Thus, for under-saturated air, $\sigma < 0$ and there is mass loss from the ice sphere (i.e. $S(D)$ is negative) and vice-versa for over-saturation.





To obtain an expression for the total mass lost or gained from phase changes, $S_q$, $S(D)$ is integrated over the number distribution and using Eqs. (5), (6) and (12), we get,

$$S_q = \int A D \sigma f_v n(D) dD, \tag{13a}$$

$$= \left\{ \frac{0.78 A \alpha}{\lambda} + C \frac{\Gamma(1 + 1.5 b_m + \alpha)}{\Gamma(\alpha) \lambda^{1 + 1.5 b_m}} \right\} N_{bs}, \text{ where } C = 0.308 A (Sc)^{1/3} a_m^{1/2} \phi^{b_m/2}. \tag{13b}$$

In a double-moment scheme there is no clear way of obtaining $S_N$ as opposed to bin-resolved schemes, where sublimation or deposition could be carried out for each bin class separately. We resort to adopting the methodology of Morrison and Grabowski (2008) to caculate $S_N$ as,

$$S_N = S_q \frac{N_{bs}}{q_{bs}} \tag{14}$$

All the quantities required to solve Eqs. (1) have now been discussed except the boundary conditions. For the upper boundary we impose a zero flux upper boundary. Considering that the top of the WRF domain would typically be greater than 10 km above the surface, the upper boundary is unlikely to have a significant impact on blowing snow dynamics. On the other hand, the lower boundary conditions (LBC) for $q_{bs}$ and $N_{bs}$ are of critical importance.

As mentioned earlier, the blowing snow model relies upon a parameterization of saltation fluxes to act as a LBC. The first step is calculating the threshold friction velocity, above which erosion of the snowpack is possible. The threshold friction velocity is directly dependent on microstructural properties of the snow surface: grain size, bond size, coordination number and sphericity of the snow grains. One of the main motivations for coupling SNOWPACK to WRF was that these properties are prognostic variables of SNOWPACK and thus available directly in CRYOWRF.

The threshold friction velocity $u_{*,t}$ is computed following Lehning et al. (2000) and Schmidt (1980) as,

$$u_{*,t} = \sqrt{\frac{A_t \rho_{ice} g r_g (SP + 1) + B_t \sigma_{ref} \frac{N_3 r_b^2}{r_g^2}}{\rho_{air}}}, \tag{15}$$

where $A_t$ and $B_t$ are geometrical parameters, SP is the sphericity of the snow grains that varies between 0 and 1, $N_3$ is the coordination number of the snow grains and $\sigma_{ref}$ is the reference shear strength (which is fixed at 300 Pa). $r_g$ and $r_b$ are the grain radius and bond size between grains, respectively. Whenever the surface friction velocity is greater than $u_{*,t}$, snow grains begin to saltate over the surface. The mass flux in saltation $Q_{salt}$ $(kg\,m^{-1}\,s^{-1})$ is computed based on either the expression by Sørensen (1991) as

$$Q_{salt} = 0.0014 \rho_{air} u_* (u_* - u_{*,t})(u_* + 7.6 u_{*,t} + 205), \tag{16}$$

or the modified version of the above expression by Sorensen (2004); Vionnet et al. (2014) as

$$Q_{salt} = \frac{\rho_{air} u_*^3}{g} \left(1 - \mathcal{V}^{-2}\right) \left[a + b\mathcal{V}^{-2} + c\mathcal{V}^{-1}\right], \tag{17}$$

where $(a, b, c)$ are three empirical parameters chosen to be as $(2.6, 2.5, 2)$ respectively (Vionnet et al., 2014) and $\mathcal{V}$ is surface friction velocity normalized by the threshold friction velocity $(u_*/u_{*,t})$. SNOWPACK includes the more accurate model (but





computationally expensive) of Doorschot and Lehning (2002) that can also be used in CRYOWRF. In this model, test particles

are launched from the surface and their trajectories are explicitly solved for. The interaction between particles and the air is

calculated using drag laws along with explicit modelling of splash and rebound mechanisms between particles in the air and

the granular snow surface. The saltation mass flux is then calculated directly from the particles' calculated motion.

To compute LBCs for $q_{bs}$ and $N_{bs}$ further steps are needed. The physical reasoning is that – and this is largely presumed –

turbulent eddies are able to pick up particles only at the top of the saltation layer. To do so, first, the height of the saltation layer

($h_{salt}$) is computed. There are two possibilities for this in CRYOWRF,

$$h_{salt} = 0.0843 u_*^{1.27} \text{, or} \tag{18a}$$

$$= z_0 + \frac{(u_e \cos \alpha_e)^2}{4 g} \tag{18b}$$

The two equations above come from Pomeroy and Gray (1990) and Lehning et al. (2008) respectively. Equation (18)a is a

purely empirical construct whereas Eq. (18)b is derived directly from splash laws for saltation, represented by $u_e = 3.1 u_*$

and $\alpha_e = 25^o$, which are the ejection velocity and angle for snow/ice particles leaving the surface respectively (Doorschot and

Lehning, 2002). The remaining terms in Eq. (18)b are roughness length ($z_0$) and acceleration due to gravity ($g$). Next, it is

assumed that the mixing ratio of blowing snow in the saltation layer decays exponentially with height ($z$) as

$$q_{salt}(z) = \frac{1}{\rho_{air}} C_{salt} \frac{\lambda_{salt} g}{u_*^2} e^{\left(-\frac{\lambda_{salt} z g}{u_*^2}\right)}, \quad \text{where} \tag{19a}$$

$$C_{salt} = Q_{salt} / u_{particle} . \tag{19b}$$

In the above equation, the saltation mass flux, $Q_{salt}$, is split into a particle velocity component ($u_{particle}$) and a concentration

component $C_{salt}$. $u_{particle}$ is diagnosed using the state of the snow surface, represented by the threshold friction velocity as

$u_{particle} = 2.8 u_{*,t}$ (Pomeroy and Gray, 1990). The decay coefficient, $\lambda_{salt}$, is chosen to be equal to 0.45 following Nishimura

and Hunt (2000). Note that if using the Doorschot and Lehning (2002) model for saltation, the mass flux at different heights

$q_{salt}(z)$ is directly computed as a part of the model and thus Eqs. (19) are not required. Thus, the LBC for blowing snow mass-

mixing ratio, $q_{bs,LBC}$ is computed as $q_{salt}(z = h_{salt})$. Assuming a mean particle diameter and using Eq. (3), $N_{bs,LBC}$ can be

found in a straightforward manner. In the current version of CRYOWRF, the mean particle diamater in saltation is chosen to

be 200 $\mu m$.

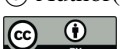



### 2.3.1 Numerics

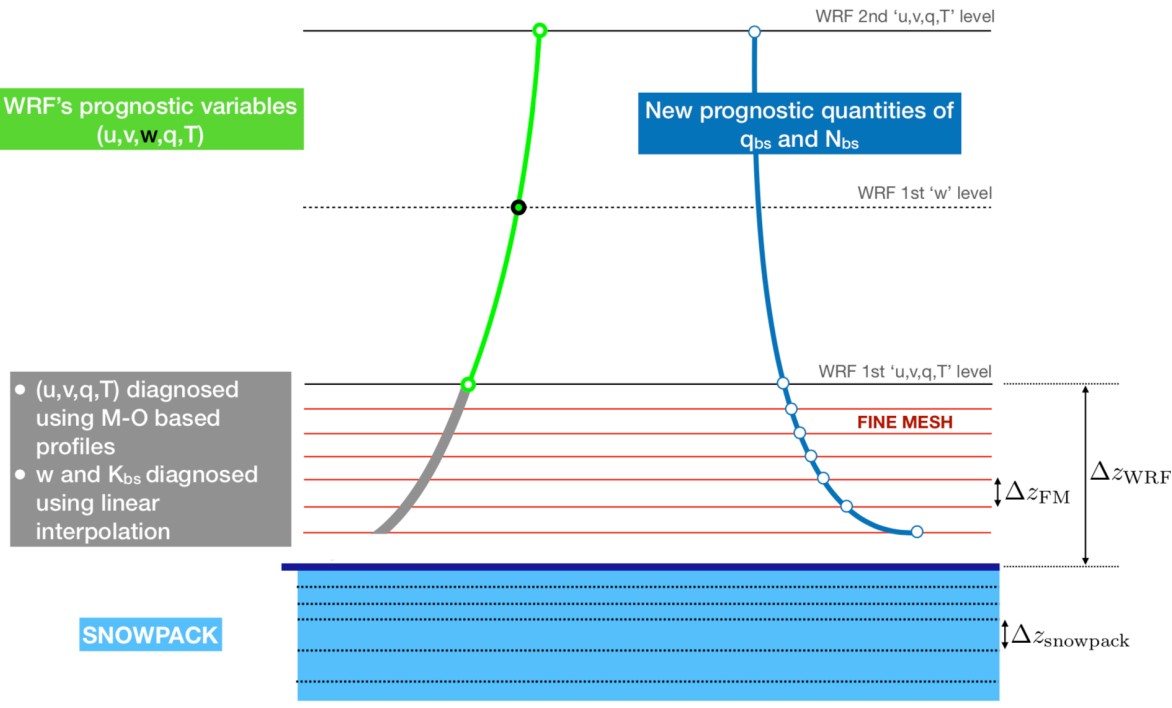

**Figure 1.** Vertical layering of the atmosphere and the snowpack: WRF's vertical mesh is staggered with vertical velocity (w) solved at mid-points (dotted black line) between levels of the other prognostic variables (solid black line) such as horizontal velocities (u,v), temperature (T) and water mixing ratios (q). Numerical nodes lying on the solid black line are also called as 'mass-points'. The new prognostic variables for blowing snow ($q_{bs}$, $N_{bs}$, blue profile) are solved on mass-points and additionally on a newly introduced fine mesh (in red). On the fine mesh, the prognostic variables of (u, v, q, T) are diagnosed using Monin-Obukhov similarity theory (grey section of the green vertical profiles). Vertical velocity (w) and the turbulent diffusion coeficcient ($K_{bs}$) are linearly interpolated between the surface and 1st 'w' level. $\Delta z_{FM}$ indicates the vertical layer thickness of the fine mesh (FM), $\Delta z_{WRF}$ indicates the layer thickness of the first WRF model level and $\Delta z_{SNOWPACK}$ shows the layer thickness of the SNOWPACK model layers.

In the previous section, all components and parametrizations that constitute the system of equations for blowing snow were
described in detail. In this section, details of the numerical algorithm to solve Eqs. (1) in CRYOWRF are explained. The designing of the numerical solver was guided (and in some ways, constrained) by WRF's overall software framework in general and in particular, the implementation of microphysics schemes. It was noted earlier that Eqs. (1) are quite similar to equations for hydrometeors such as cloud ice and thus, this approach is in many ways, justified by the solvers for those species. As described earlier, the prognostic equations of $q_{bs}$ and $N_{bs}$ are of the advection-diffusion type with additional terms for
sedimentation and phase change. Describing the algorithm in terms of the mass-mixing ratio $q_{bs}$ suffices since it is similar for $N_{bs}$.





The equation for $q_{bs}$ is spatially discretized to cell centers of the Arakawa C-grid, referred to as *mass points* in WRF's terminology. These are the same grid points where various prognostic scalars such as temperature, moisture and other hydrometeors are defined. The vertical discretization is illustrated in Fig. 1. As shown in this figure, there is an additional fine-resolution mesh

uniquely for the blowing snow variables between the surface and the first WRF *mass-point* level. The additional fine mesh (FM) is required because the blowing snow variables have extremely large gradients in the vertical as compared to other scalars (e.g. temperature), because of the impact of sedimentation. In mesoscale models such as WRF, the first model level above the surface, $\Delta z_{WRF}$ is $\sim \mathcal{O}$ (10 m), which is far too coarse to capture the sharp gradients near the surface. The fine mesh has a mesh size, $\Delta z_{FM}$ is $\sim \mathcal{O}$ (1 m) with the first FM level fixed to 50 cm above the surface. The number of levels in the fine mesh can be

chosen at run-time and remains fixed during the simulation. Typically, 8 to 10 levels are sufficient to capture the near-surface gradients. Values for WRF's prognostic variables are diagnosed for points on the fine mesh using either the stability-corrected log laws (u, v, T, q) or simple linear interpolation (w and $K_{bs}$). The implementation of the FM in CRYOWRF is inspired by SURFEX (Vionnet et al., 2012). In SURFEX, there is a prognostic, multi-layer *surface layer* scheme called CANOPY (Masson and Seity, 2009) whereas in the current version of CRYOWRF, the values of the atmospheric variables are only interpolated

onto the fine mesh.

To follow WRF's existing framework to solve for hydrometeors, we begin by reformulating Eq. (1a) as,

$$\frac{\partial q_{bs}}{\partial t} = \left[\frac{\partial q_{bs}}{\partial t}\right]_{m+s} + \left[\frac{\partial q_{bs}}{\partial t}\right]_{p} + \left[\frac{\partial q_{bs}}{\partial t}\right]_{advec}, \tag{20}$$

where, the three terms from left to right in the R.H.S of the above equation are contributions from turbulent mixing and sedimention (m+s), phase change (p) and advection (advec) respectively. They are stated as

$$\left[\frac{\partial q_{bs}}{\partial t}\right]_{m+s} = -K_{bs}\frac{\partial^2 q_{bs}}{\partial x_i^2}\delta_{i3} + \frac{\partial}{\partial x_i}\left(q_{bs}V_q\delta_{i3}\right) \tag{21a}$$

$$\left[\frac{\partial q_{bs}}{\partial t}\right]_{p} = S_q \tag{21b}$$

$$\left[\frac{\partial q_{bs}}{\partial t}\right]_{advec} = -u_i\frac{\partial q_{bs}}{\partial x_i} \tag{21c}$$

Equation (21a) is solved in a semi-implicit fashion (fully implicit for turbulent mixing and explicit for sedimentation) using second-order finite differencing, resulting in a tri-diagonal matrix that is solved using the well-known Thomas Algorithm.

The semi-implicit matrix-based solver is required to make the solver numerically stable and be able to deal with the fine resolution of the fine mesh and the relatively coarse resolution of WRF's native mesh in the same matrix. Immediately after, Eq. (21b) is solved in an explicit fashion for both the fine mesh as well as WRF's native mesh together. The heat and mass transfer quantities in the region of the fine mesh are added as correction terms to the first WRF level to ensure conservation of mass and energy. Equation (21c) is solved using WRF's existing advection routines for scalars, similar to how hydrometeors

are advected. The possible advection algorithms in WRF range from finite difference schemes of different orders, positive definite schemes, WENO type schemes etc. Temporal integration of Eq. (21c) is done using 2nd or 3rd order Runge Kutta. The numerical algorithms for advection of blowing snow quantites is, at present, constrained to be the same as that for other scalars.





Finally, the mass balance at the surface due to precipitation and snow transport, $\Delta m_{sfc}\,[\mathrm{kg\,m^{-2}\,s^{-1}}]$ is computed as

$$\Delta m_{sfc} = P + \int_0^\infty \left[\frac{\partial q_{bs}}{\partial t}\right]_{m+s} \rho_{air}dz + \nabla_H Q_{salt} \,,\text{where},\tag{22a}$$

$$\nabla_H = \frac{1}{s}\left(\frac{\partial u_{WRF}}{\partial x} + \frac{\partial v_{WRF}}{\partial y}\right) \,,\, s = \sqrt{u_{WRF}^2 + v_{WRF}^2}\tag{22b}$$

The first term on the R.H.S of the above equation is precipitation (P). The second term is the vertically integrated L.H.S of Eq. (21a) and represents the contribution of snow transport, specifically snow suspension. The final term, the horizontal divergence of saltation flux, $Q_{salt}$ is the contribution of saltation to the surface mass balance. It is computed using the horizontal wind components at the first level of WRF $(u_{WRF}, v_{WRF})$ above the surface. The above formulation for surface mass balance is similar to that implemented in ALPINE3D (Lehning et al., 2008).

### 2.4 Coupling Library

A special coupling library was implemented to link SNOWPACK and WRF. This was necessitated by the fact that SNOWPACK is a C++ codebase and WRF, especially the physics routines, are largely written in FORTRAN. The coupling library has twin tasks. Firstly, the coupler acts as an API that exposes SNOWPACK's C++ routines and makes them callable from FORTRAN codes. Secondly, as SNOWPACK follows OOP principles quite strongly, the coupling library allocates and maintains pointers to SNOWPACK objects during runtime.

The coupling library is written in a mixture of C++, C and FORTRAN and consists of, for all practical purposes, two FORTRAN-callable subroutines, one for initialization and one for a single time-step execution. Thus, WRF maintains the timestepping control, as is commonly the case for the interaction between an atmospheric model and the land surface model. Domain decomposition (MPI) and OpenMP based loop parallelization is also organized by WRF's software framework and therefore all computations within SNOWPACK's objects are serially executed.

The main design target for the coupling library was to ensure that it is agnostic of both the WRF and SNOWPACK versions. This is important because both codebases are under active development and new versions with new or improved physical parameterizations or bug-fixes are regularly released. The design target promises a long shelf-life and thus ensures the viability of CRYOWRF in the future.

## 3 Model capabilities and performance

The overarching goal, while developing CRYOWRF from a usability perspective, was to make its adoption extremely simple for both new and experienced WRF users. This means that the entire workflow, beginning from the compilation of the model, the pre-processing, the actual simulation to even post-processing of the results is ensured to be compatible with WRF, with only very minor changes from the standard process. Significant effort was also expended to ensure that CRYOWRF adopts all the capabilities of the parent WRF and SNOWPACK models. We list the most relevant of these capabilities below.



– Initialization: the snowpack is initialized through per-pixel ascii files in the *.sno format. (This format is a standard format for SNOWPACK. For more details see, https://models.slf.ch/docserver/snowpack/html/snowpackio.html). Helper Python scripts are provided to generate these files from the *wrfinput* files generated by running the WRF pre-processing program *real.exe* that is a pre-requisite for running WRF.

– Restart: Once the model simulation has started, restarting the simulation (this is quite common for long simulations) is exactly the same as the native WRF simulations. This is because all the relevant snowpack details, including depth profiles of snowpack quantities (snow temperatures, density, snow grain size etc.) are stored in the *wrfrst* files.

– IO: All snowpack outputs are managed by WRF's IO streams. This includes depth profiles but also additional diagnostics such as terms for mass and surface energy balance.

– Nesting: CRYOWRF allows flexible launching of inner, high resolution nests. This is done by performing nearest-neighbour interpolation to initialize snowpacks in the inner nest from the coarser outer nest.

An additional capability is that the timestep of SNOWPACK ($\Delta t_{SN}$) is flexible and can be any integer multiple of WRF's timestep ($\Delta t_{WRF}$), as illustrated in Fig. 2. It is important to note that ice mass ( blowing snow and precipitation) is exchanged between WRF and SNOWPACK at every $\Delta t_{WRF}$. This is particularly important to account for erosion/deposition due to blowing snow as the blowing snow model is necessarily calculated at every WRF timestep. At every ($\Delta t_{SN}$), the heat equation for snowpack, along with metamorphism, settling and melt-freeze of the snowpack layers are solved. The forcings from WRF are the incoming shortwave and longwave radiation along with temperature, relative humidity and wind speed and SNOWPACK returns to WRF, the sensible and latent fluxes along with albedo. The flexible timestepping capability can be quite important as, depending on the settings used for SNOWPACK, many tens to hundreds of snow layers may have to be accounted for. For such cases, calling SNOWPACK every timestep would be prohibitively expensive without significantly increasing the accuracy of the simulation from the atmosphere's perspective.

The computational costs of using SNOWPACK as a LSM scheme in relation to existing LSM models were assessed. A simulation domain with 64×64 surface pixels was chosen (it was ensured that each pixel in the domain is a *land* pixel and no *water* pixels are found in the domain). A baseline simulation was performed using Noah-MP along with CRYOWRF simulations with 10, 50, 100 and 400 layers of snow. The number of layers was kept fixed during the course of these benchmarking simulations. For each CRYOWRF setup, three variants were simulated. These were (i) with WRF and SNOWPACK timesteps being equal ($\Delta t_{SN} = \Delta t_{WRF}$),(ii) timestep of SNOWPACK being ten times that of WRF ($\Delta t_{SN} = 10\Delta t_{WRF}$) and finally (iii) same as the previous variant but with the blowing snow module switched off. An additional simulation with the Community Land Model (CLM) was also performed. Apart from changes in the LSM setup, all the simulations had exactly the same setup (WRF timestep, radiation, boundary-layer scheme, etc) and were compiled in exactly the same compiler and compiler settings.

The results of these simulations are presented in Fig. 3, with percentage increase in run-time with respect to the Noah-MP baseline case chosen as the metric. For reference, it must be noted that Noah-MP and CLM use 3 and 5 snow layers respectively. The first variant, with equal SNOWPACK and WRF timesteps, scales (nearly linearly) with number of simulated





snow layers with a 30 % increase for 10 snow layers and increasing up to 240 % increase with using 400 snow layers. From this perspective there is indeed a significant overhead of using SNOWPACK as the LSM. The benefits of variable timestepping for SNOWPACK are immediately clear when comparing the costs of the second variant. In this case, while there is an increase in run-time, it is not as significant, with a much lower, 38 % increase for 400 snow layers. For simulations with more *reasonable*
numbers of snow layers, the computational costs are less than 15 % higher than Noah-MP. To put SNOWPACK's computational overhead in perspective, the CLM model (being called every timestep), requires 84 % longer runtime. The overhead due to the blowing snow module is insignificant in all cases. This analysis proves that using an advanced-complexity snow model coupled to WRF is feasible from a performance perspective.

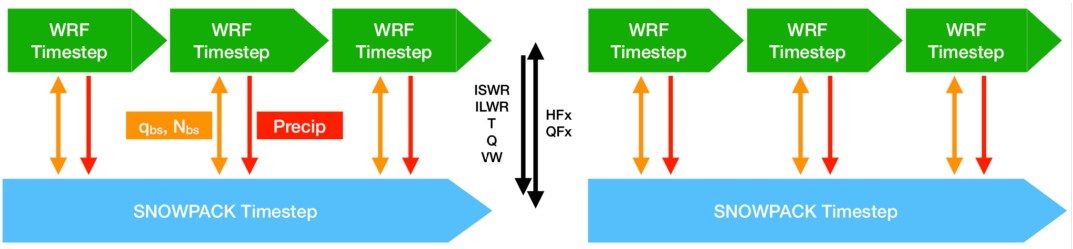

**Figure 2.** Flexible time-stepping of SNOWPACK within CRYOWRF: SNOWPACK can be called at any integer multiple of WRF's timestep. In this schematic, SNOWPACK is called every 3 WRF timesteps. The orange (blowing snow) and red (precipitation) arrows indicate *fast* exchange of mass between the snowpack and WRF. At intervals of $\Delta t_{SN}$, *slow* dynamics for thermal processes are accounted for by solving the heat equation in the snowpack.

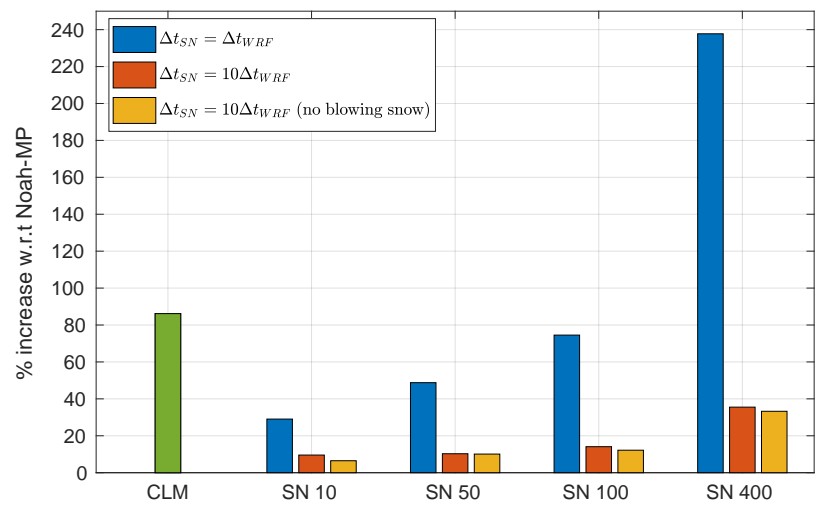

**Figure 3.** Runtime cost of CRYOWRF: Percentage increase in runtime due to increasing number of simulated snow layers. Number of snow layers simulated are 10, 50, 100 and 400 layers. The percentage increase is calculated with respect to a simulation with Noah-MP being treated as a baseline. An additional comparison with CLM model is also shown. Different colored bars represent the computational overhead without (blue bars) and with SNOWPACK's variable timestep (red bars) along with the overhead of simulating blowing snow (yellow bars).





# 4 Case Studies using CRYOWRF

In this section, several case studies using CRYOWRF are presented. These case studies are intended primarily to showcase the breadth of scenarios that CRYOWRF can simulate at present. These scenarios range from continent-scale simulations covering the entirety of Antarctica for climate relevant studies to local, valley scale simulation in Alps. Note that the presentation of the case studies does not go into detailed intercomparisons with measurements etc. This is out of scope for this article and would be done in dedicated studies in the future. Thus, in the first case-study, intercomparison with data is presented to provide

evidence of the model's accuracy, while the remaining results are presented directly from simulation outputs.

## 4.1 Case Study Ia: Continent-scale simulation of Antarctica for mass balance

### 4.1.1 Objectives

The first case study is a one year (July 2010-July 2011) simulation for the entire continent of Antarctica using CRYOWRF. The chosen simulation setup is standard for mass balance calculations in Antarctica and comparable with the most common models

being RACMO and MAR (e.g. Agosta et al., 2019). The intention of this case study is to demonstrate the accuracy of the CRYOWRF model along with showcasing the kind of information now available directly from CRYOWRF, particularly with respect to the snow-cover. This, as well as the following case studies, also serve as tutorials to ease the adoption of CRYOWRF for interested researchers.

### 4.1.2 Simulation Setup

The simulation is performed over a single domain with a horizontal resolution of 27 km, covering the domain shown in Fig. 4. The domain encompasses the entire continent along with a substantial portion of the surrounding Southern Ocean. Static data (topography and landuse) are from the 1-km resolution Reference Elevation Model of Antarctica dataset (REMA Howat et al., 2019) and the AntarcticaLC2000 landuse data (Hui et al., 2017), respectively, and were prepared as WRF input by Gerber and Lehning (2020). The model is initialized and forced at the lateral boundaries by ERA5 reanalysis data (Hersbach et al.,

2020). The parameterizations used are: RRTMG (shortwave and longwave radiation), Kain-Fritsch (KF, convection), a modified version of the Morrison scheme that improves representation of secondary ice production (microphysics, Vignon et al., 2021; Sotiropoulou et al., 2021), and MYNN (boundary layer scheme). The simulation is nudged against ERA5 for wind speeds in zonal and meridional direction in the top 20 levels of the atmosphere.

The land surface model (LSM) is the SNOWPACK model. The snow cover is initialized using data from Utrecht University's

Firn Densification Model (FDM, Ligtenberg et al., 2011) forced by RACMO 2.3 (van Wessem et al., 2018), which provides vertical profiles for snow layer thickness, density and temperature along with grain radius for the entire continent. Pixel-wise ASCII files are created from the FDM data (which is in netcdf format). The simulation is initialized for only the top 20 meters of the snowpack. Snow cover modelling requires parametrizations for many different processes. To calculate albedo, we used the model developed by Munneke et al. (2011) and implemented in SNOWPACK by Steger et al. (2017b). The new snow density



is fixed to 300 kg/m$^3$ and the water transport through a snow column is done using the bucket approach. Snow-atmosphere exchange of mass and energy is calculated using the Holtslag surface layer model (Schlogl et al., 2017). The snow surface roughness is fixed at 0.002 m.

The SNOWPACK model allows for *smart* merging (splitting) of vertical layers based on their similarity (strong gradients), apart from the physical processes of snow accumulation, melting and settling. The minimum thickness of a snow layer is set

to 0.001 m. This is indeed quite a small value and was chosen as such to simply demonstrate CRYOWRF's ability to model snowpacks at such *extreme* resolutions. The blowing snow model was active with an eight layer fine mesh between the surface and the first *mass* point of WRF. The resolution of the fine mesh extended between 0.5 m (closest to the surface) to 1.5 m.

### 4.1.3 Atmospheric weather stations

Results of this simulation are compared to weather station measurements from 14 stations spread throughout the continent

(mostly along the coast, marked in red in Fig. 4). Station data is taken from several databases. A summary is given in Table 1. No thorough quality checks were performed for the station data for this study. Data gaps are commonly occurring. Some stations provide no information about relative humidity (Mizuho), pressure (D-17) and wind direction (D-17). In addition to a verification with AWS data, figures of the surface mass balance components and snow profiles showcasing the capabilities of the entire CRYOWRF framework (including diagnostics) are presented in the following section.

| Database | Station | Reference |
|---|---|---|
| AAD | Casey | https://data.aad.gov.au/aws |
| | Davis | |
| | Mawson | |
| AMRC | Mizuho | ftp://amrc.ssec.wisc.edu/pub/aws/q1h/ |
| | D-10 | |
| | D-47 | |
| | D-85 | |
| CLIMANTARTIDE | Concordia | https://www.climantartide.it/dataaccess/index.php?lang=en |
| | Eneide | |
| IMAU | Kohnen | Lazzara et al. (2012), https://www.projects.science.uu.nl/iceclimate/aws/antarctica.ph |
| NOAA | South Pole | https://gml.noaa.gov/aftp/data/meteorology/in-situ/spo/ |
| PANGAEA | Neumayer | e.g. König-Langlo (2012) |
| Other | D-17 | Amory (2020) |
| | Princess Elisabeth | KU Leuven, Prof. N. van Lipzig, personal communication |

**Table 1.** Atmospheric weather stations used for model verification.

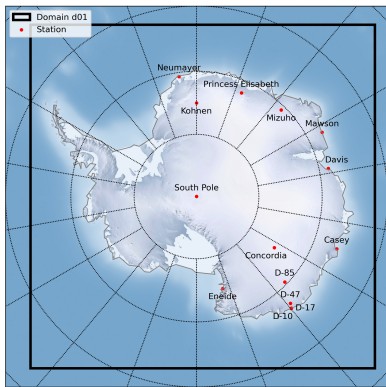

**Figure 4.** Simulation domain for Case Study Ia (black rectangle). The simulation domain encompasses the entire continent. Additionally, measurement stations used for model verification are represented by red dots.

### 4.1.4   Results

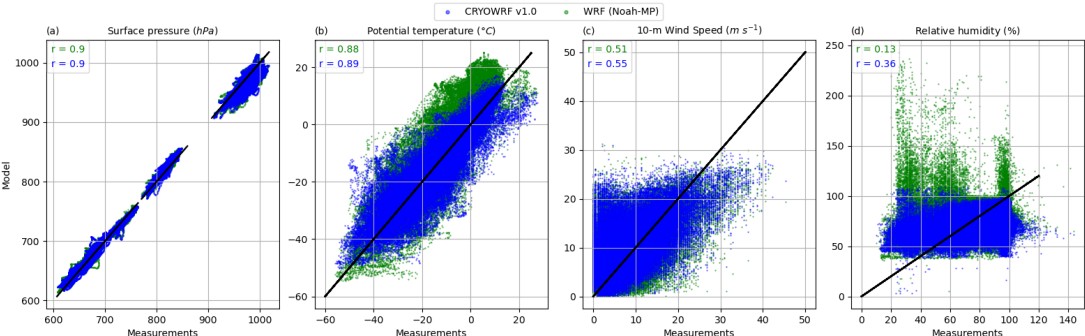

**Figure 5.** Correlation plots between CRYOWRF timeseries output (blue) and measurements. All measurements are consolidated. For comparison, results with Noah-MP are shown (green). The correlation results are presented for surface pressure (a), 2 meter potential temperature (b), 10 meter wind speed (c) and 2 meter relative humidity (d).

Fig. 5 shows scatter plots comparing the measurements (on the horizontal axes) with CRYOWRF outputs (on the vertical axes) for the four standard meteorological variables surface pressure, potential temperature and relative humidity at 2 meters above surface and wind speed at 10 meters above surface. For comparison, results from a simulation with Noah-MP as the LSM are also shown. The best correlation is found for surface pressure (r=0.90) and the worst correlation for relative humidity (r=0.31) with CRYOWRF outperforming or similar to Noah-MP for each of the variables, though admittedly, the improvements over Noah-MP are minor except for relative humidity.

South Pole was chosen to present an example of direct comparison of timeseries between CRYOWRF, Noah-MP and measurements. These are shown in Fig. 6. In addition to timeseries, a comparison of the probability density functions (PDF) for the entire timeseries is shown in adjoining figures on the right for each row. In addition to the four variables presented earlier, wind

direction timeseries are shown as well. It is clear that CRYOWRF follows the measurements reasonably well. Additionally, CRYOWRF performs better than Noah-MP with lower mean abolute errors (mae), for each variable. The best improvement is found for potential temperature and relatively humidity. The same conclusions can be drawn from the PDF plots. For example, the PDFs of CRYOWRF and measurements of potential temperature have a much between overlap in comparison to that with Noah-MP, which has a significant warm bias.

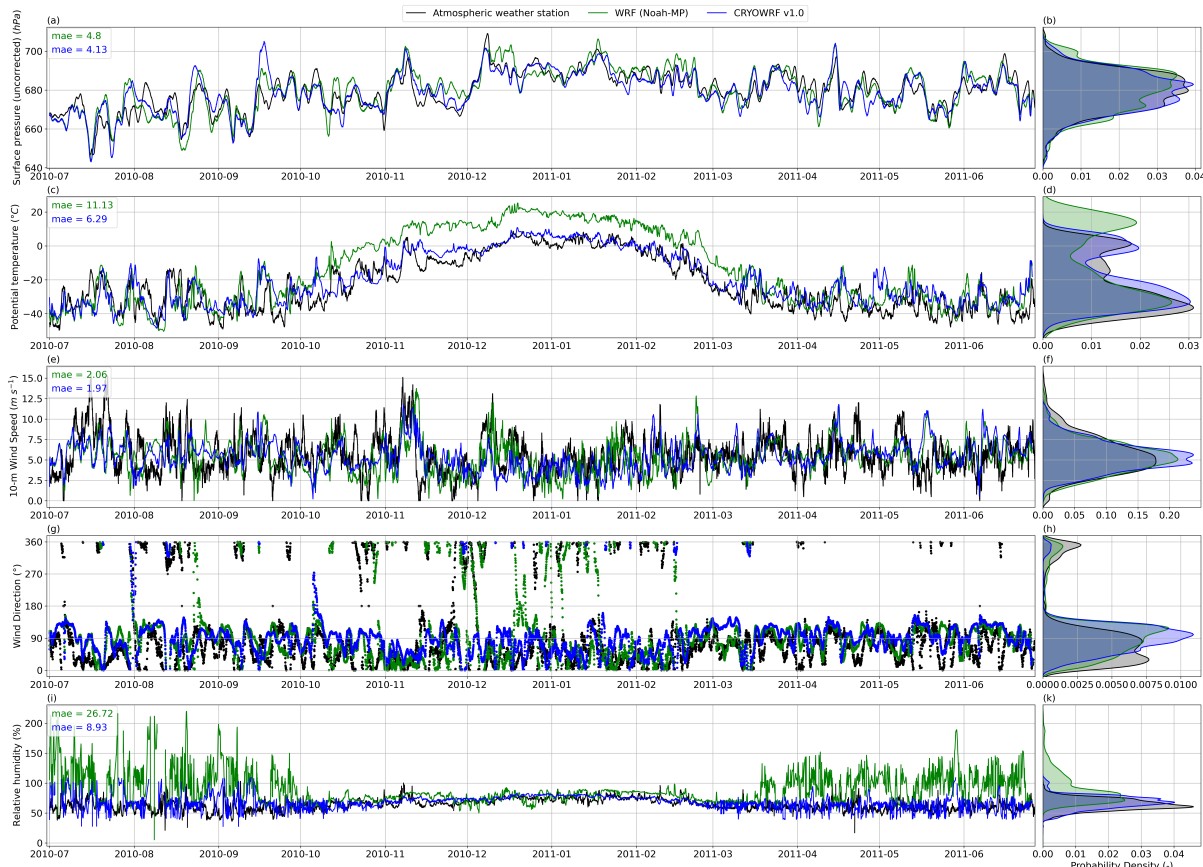

**Figure 6.** Timeseries comparison between CRYOWRF, Noah-MP and measurements using the automatic weather station at the South Pole for surface pressure (a), 2 meter potential temperature (c), 10 meter wind speed (e), wind direction (g) and relative humidity (i). The distributions of these quantities over a period of one year are shown in the adjoining figure on the right in each row (b, d, f, h, k).

One major reason for performing simulations such as in this case study are to calculate components of the surface mass balance. The polar surface mass balance is of critical importance in understanding the implications of global warming on sea-level rise. CRYOWRF outputs these quantities as two-dimensional fields directly as a part of the standard output via its specially implemented diagnostics package. An example of the surface mass balance output during this case-study is shown in Fig.7. Since the simulation period is of only one year, it is not possible to quantitatively compare the results presented in this

figure with those published in literature, which are typically calculated over climate time scales of at least 30 years. However,





qualitatively, the spatial patterns for each of the different terms matches quite well (Agosta et al., 2019, for example). For example, precipitation is dominant in West Antarctica and the Peninsula while in East Antarctica, it is limited to the coasts with strong gradients along the topographic contours. It is interesting to see melt-refreeze dynamics being dominant in the Ross and Ronne ice shelves. Submilation (or vapor deposition) patterns reveal the importance of this mechanism on the coastline

while identifying locations in the interior of the continent where vapor deposition is actually a net positive term in the surface mass balance.

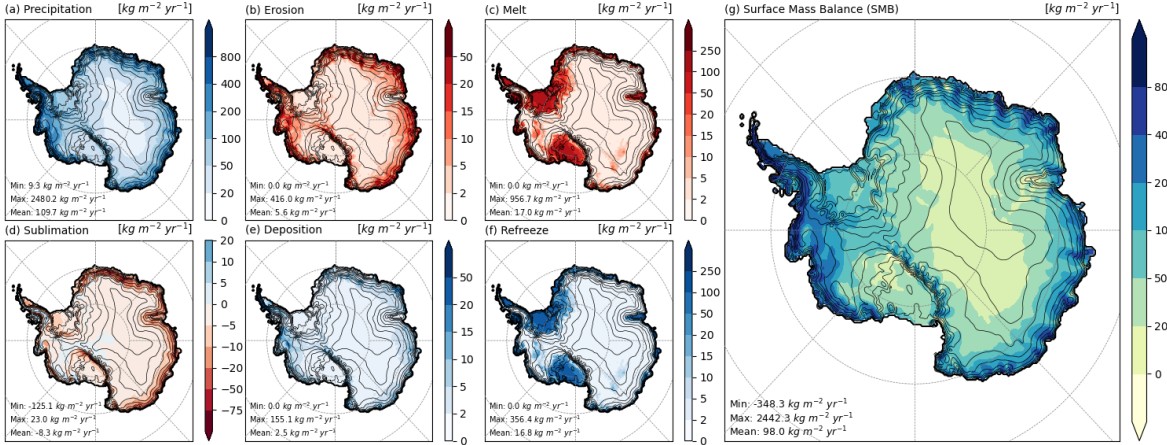

**Figure 7.** Components of the surface mass balance output directly from CRYOWRF.



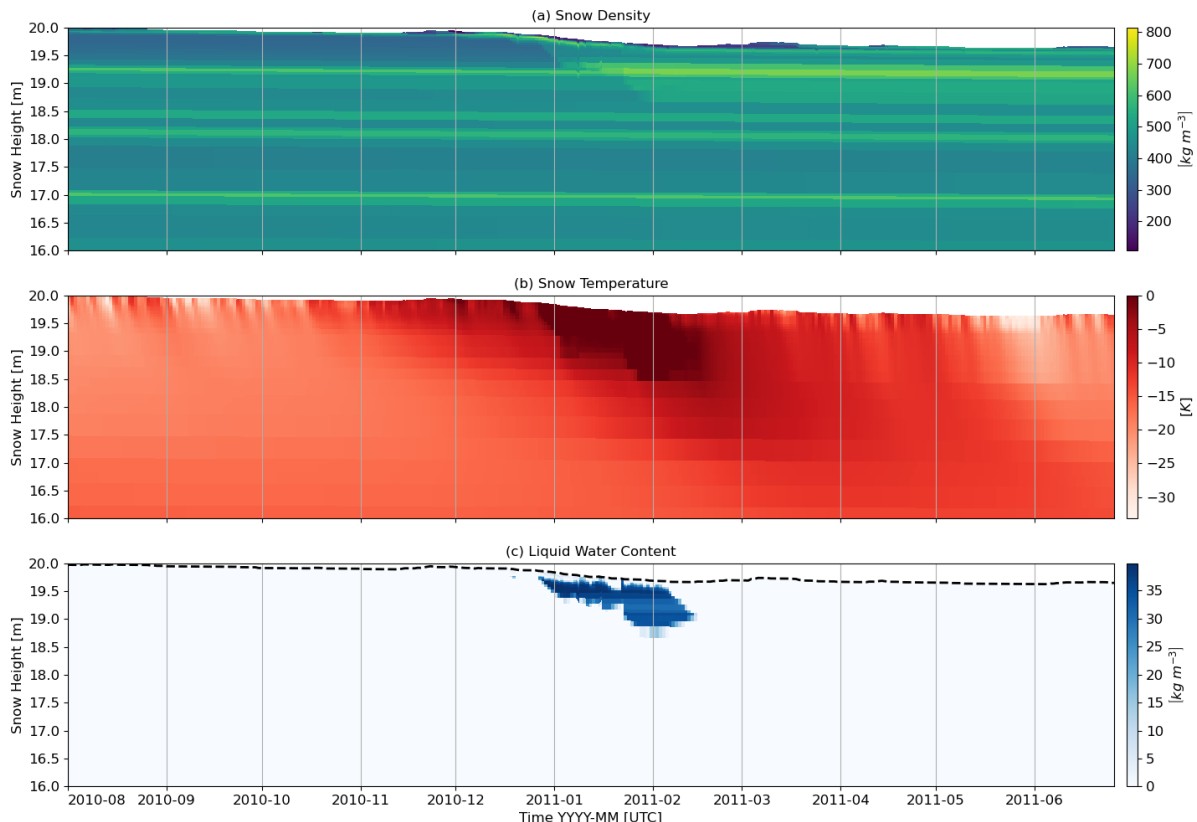

**Figure 8.** Profiles of snow density (a), temperature (b) and liquid water content (c) at the Casey station (66.4058 S, 110.8043 E) simulated by CRYOWRF for the duration of the one year simulation between August 2010-June 2011.

A notable feature of CRYOWRF is its ability to run atmospheric simulations with highly (vertically) resolved snowpacks. This allows accurately representing thin subsurface ice layers of thickness of the order of a few millimeters that have significant impact on snow hydrology at larger scales (Wever et al., 2016b). In Fig. 8, the evolution of the snowpack at the Casey station

(66.4058 S, 110.8043 E) on the East Antarctic coastline is represented by the density, temperature and liquid water content (LWC) profiles. For this location, based on the local mass and thermal forcing, SNOWPACK converges to 130 snow layers. As can be seen in the density profiles, thin layers of high density snow (or ice) layers can be found within the top one meter of the snowpack. These layers are mostly less than a centimeter thick and are between layers of (in some cases, considerably) lower density. The top one meter of the snowpack is also thermally active with significant cooling/warming at the diurnal and seasonal

scale, relative to sub-surface temperature fluctuations. The exceptional period is period between the middle of December till the middle of February when significant melt occurs, reducing nearly 0.5 m of snow. The melt water percolates through the





snowpack up to a depth of one meter resulting in isothermal, melting point temperatures at the beginning of February. The meltwater percolation is clearly seen in the LWC profiles where, significant meltwater is found to remain in a localized region at the depth of 1 meter until the middle of February.

## 4.2 Case Study Ib: Multiscale simulation at DDU with blowing snow: formation of hydraulic jumps

### 4.2.1 Objectives

The second case study simulates the formation of an hydraulic jump at the coast surrounding the Dumont d'Urville station between 8th and 11th of August, 2017. The hydraulic jump occurs when a well-developed, high velocity katabatic flow draining off the sloped ice sheet of Antarctica reaches the coast. The high velocity outflow experiences an abrupt and rapid transition due to changes in togographic and surface conditions thereby transitioning to a hydraulic jump. The hydraulic jump itself further triggers gravity waves. A remarkable manifestation of the hydraulic jump, given the 'right' surface conditions, is the large-scale entrainment and convergence of blowing snow particles within the hydraulic jump. This can result in formation of 100-1000 m high, highly localized 'walls' of snow in the air in an otherwise cloud-free sky.

The case study is directly inspired by the recent work of Vignon et al. (2020) who present a detailed analysis on the genesis, dynamics and implications of the hydraulic jumps and the corresponding gravity waves in the region in proximity of DDU. Their study used (standard) WRF as the principal tool and showed that capturing this phenomenon in simulations is only possible with high-resolution domains with resolutions higher that 3 km at the very least.

This case study serves as a perfect example to highlight the capabilities of CRYOWRF both from a physical modelling as well as a technical perspective. CRYOWRF, due to its non-hydrostatic atmospheric core can indeed simulate such phenomenon, in direct contrast with the pre-eminent mesoscale models for polar research, namely, RACMO and MAR. From a technical perspective, performing such a simulation necessarily requires a nested domain approach. As mentioned earlier, CRYOWRF is compatible with the nesting capability of WRF. Finally, unlike, Vignon et al. (2020), who used standard WRF, CRYOWRF includes the blowing snow model, which allows for studying the impact of hydraulic jump formation on blowing snow dynamics and simulate the formation of the aforementioned 'snow walls' for the first time.

### 4.2.2 Simulation Setup

The simulation setup is as follows: A three-domain simulation is performed having horizontal resolutions of 12 km (d01), 4 km (d02) and 1km (d03) for the three domains. All the domains are centered on DDU with the Adelie Land coastline approximately passing diagonally across the domains. The domain locations and extents are shown in Fig. 9. The nesting is of a 'one-way' nature, i.e., the inner, higher resolution domain is forced at the lateral boundaries by the outer, coarser domain, while there is no upscaling of the inner domain data to nudge the outer domain. The outer-most domain (d01) has exactly the same setup as that in the previous case study (apart from the resolution) – the same static data and physical parametrizations are used and ERA-5 is used to force the simulations. The two inner domains also have the same setup as the outer domain with the exception that the convection parametrization is switched off. The blowing snow model is active in all three domains.





The simulation starts with only the outer-most domain first, with the inner domains activated successively. The snow profile
525    at each of the pixels is initialized using *.sno files generated from a restart (*wrfrst*) file from an coarse resolution simulation,
similar to that in Case Study Ia. The starting times for the three domains are the 8th of August 2017 at 0000 hrs UTC, 0600
hrs UTC and 9th of August 2017 at 0000 hrs for the three domains respectively. The simulation of all three domains continues
until 11th of August 2017. Increasing resolution implies decreasing time-steps. The three domains used timesteps of 45, 15
and 5 seconds, respectively. The LSM (SNOWPACK) uses a timestep of 900 seconds in each of the three domains, using the
530    in-built capability of CRYOWRF to call the LSM module with flexible timesteps.

In the following section, two illustrations of model output are shown where the hydraulic jump and the excited gravity
waves along with its effect on blowing snow can be seen. Differences between the domains on account of the resolution are
also highlighted.

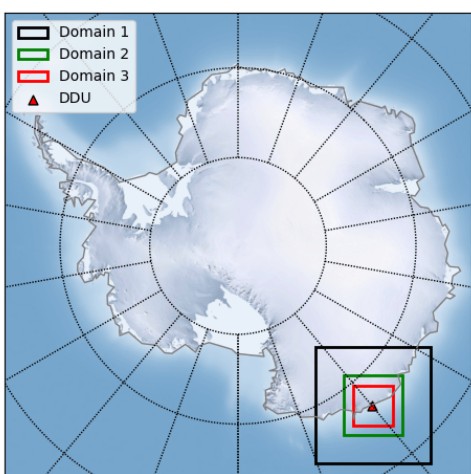

**Figure 9.** Simulation domains for Case Study Ib (colored rectangles). The three nested domains of 12 km (black), 4 km (green) and 1 km
(red) resolution. These domains are centered at the Dumont d'Urville (DDU) station (red triangle).

### 4.2.3    Results

535    As reported by Vignon et al. (2020) using station measurements, there is a relatively quiescent period before the 9th of August
2017 with stable wind speed, wind direction and air temperature recorded at DDU. The situation dramatically changes on the
9th when the katabatic flow rapidly strengthens and very high wind speeds are reported at DDU in the first half of the 10th.
At 1200 hrs on the 10th, there is a sudden drop in the wind speed, following which the wind becomes gusty but on average
remains slower than the previous day. Vignon et al. (2020) showed that the abrupt deceleration of the katabatic flow is due to
excitation of gravity waves and the enhanced friction induced.

These trends are reproduced by CRYOWRF and shown in Fig. 10, where the simulated timeseries of wind speed, direction
and temperature at the DDU station are plotted. It is interesting to note the differences in the trends shown by the three





nested domains with different resolutions, particularly for wind speeds and direction. The coarse resolution Domain 1 (12 km resolution) follows the trends mentioned earlier but the rate of wind acceleration and its abrupt slowdown are significantly less

pronounced than the finer resolution nests, which match each other quite well. Differences between the two inner nests emerge in the case of wind direction, where the flow in Domain 3 shows significant oscillations in flow direction in the period before and after 1200 UTC on the 10th of August. The flow direction resolved at 12 km resolution in Domain 1 remains relatively fixed on the other hand. The air temperature, initially quite stable on the 8th experiences a rapid decrease that coincides with the change in wind direction as well as an acceleration. This decrease continues until the middle of the 9th followed by a

warming trend. Interestingly both the inner, higher resolution domains are slightly warmer during the katabatic phase and at the onset of the gravity wave excitation, where temperature oscillations can be observed. These plot perfectly highlight the need for CRYOWRF's high resolution capability - the intensity of the katabatic jet, its sharp transition into a hydraulic jump can only be resolved by increasing the resolution from 12 km to 4 km. The further phenomenon of gravity wave excitation is only possible by increasing the resolution even further to 1 km (Vignon et al., 2020).

The impact of the flow transitions on blowing snow dynamics are illustrated in the following two figures. Figs. 11 and 12 both show blowing snow mixing ratio $q_{bs}$ [kg/kg] on the 10th of August 2017 at 1200 UTC, for each of the three domains. This time period was chosen as the onset of gravity waves occurs close to this time (based on Fig. 10). Blowing snow at 1,500 m above ground level (AGL) is shown in Fig. 11 with DDU marked on the map (red triangle) along with the coastline (black line) for reference. In all three domains, there is a boundary delineating regions with and without blowing snow. The

boundary is precisely the location of the hydraulic jump. The effect of model resolution is also clearly apparent. In the coarsest domain (Domain 1, 10a), particle concentrations are lower and more uniformly spread. Increasing the resolution from 12 km to 4 km, the concentration values increase, while at the same time, the part of the domain, where blowing snow particles exist at this height shrinks to a smaller area as compared to the outer domain. Finally, in the innermost, 1 km resolution domain, there is a clear imprint of wave-like fluctuations on the blowing snow concentration field. Vertical cross-sections of blowing

snow mixing ratio are extracted at the dashed red line passing through DDU and are shown in Fig 12. Additionally, potential temperature isotherms for values between 260 and 290 K are shown to indicate the presence of gravity waves. Similar to the previous figure, the flow in each of the three domains shows the characteristic hydraulic jump approximately 25 km inland from the coast. As expected, increasing the resolution results in a sharper localization of the jump location. Further inland from the coast, a thin region with high concentration of blowing snow particles can be observed. These particles are entrained

into the hydraulic jump and are mixed up to much greater altitude off the coast. High concentrations are found even at 2000 m above sea level (ASL). Only the innermost, 1km resolution nest exhibits signs of gravity wave excitation as evidenced by the wavy perturbations to the isotherms, particularly those for 270 K and 280 K. Correspondingly, spatial patterns of blowing snow mixing ratio matches the envelopes of the wave-trains at different altitudes quite well. A video that shows the dynamics of the blowing snow mass mixing ratio for the entire simulation period is provided as Supplementary Video M1. This video

shows the impact of the acceleration of the katabatic flow on blowing snow dynamics, the formation of the hydraulic jump and the excitation of the gravity waves in the innermost 1-km domain.



This case study requires further detained verification and analysis even though qualitatively, it reproduces satellite and visual observations of *snow walls* reported at DDU (see Vignon et al. (2020) for more data sources). More pertinent for the scope of this article is that this case-study showcases an example of how CRYOWRF can contribute to enquiry in coupled snow-
atmosphere interaction in polar regions that necessarily require high resolution. Extensive blowing snow clouds, with particles regularly reaching as high as 500 meters above the surface have been observed in satellite measurements (Palm et al., 2017) that have not yet been accurately simulated by models. High-resolution simulations of the sort presented in this case-study may be useful in matching or explaining the satellite measurements and analyses of Palm et al. (2017).

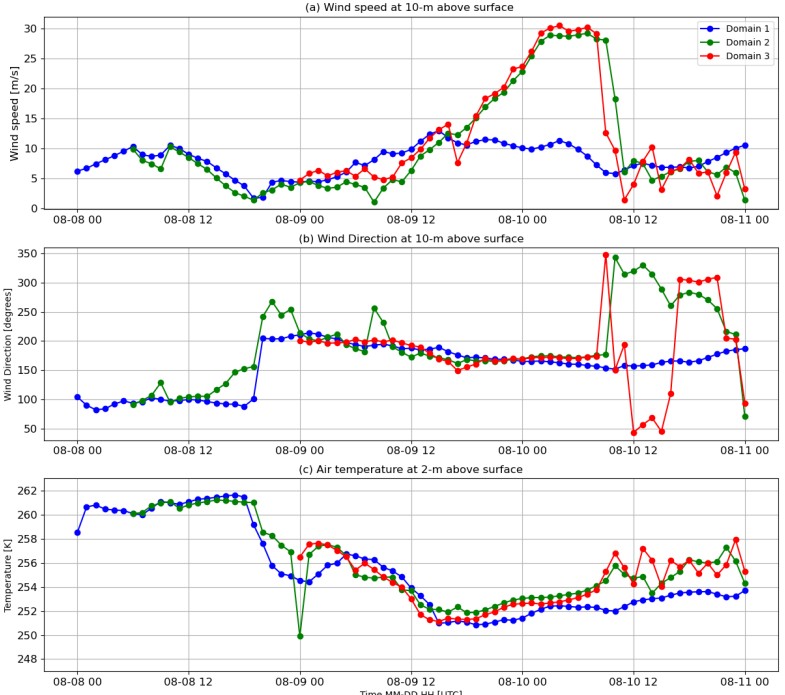

**Figure 10.** Meteorology at Dumont D'Urville as simulated by CRYOWRF. (a) Wind speed and (b) Wind direction at 10 meters above surface and (c) air temperature at 2 m above surface.



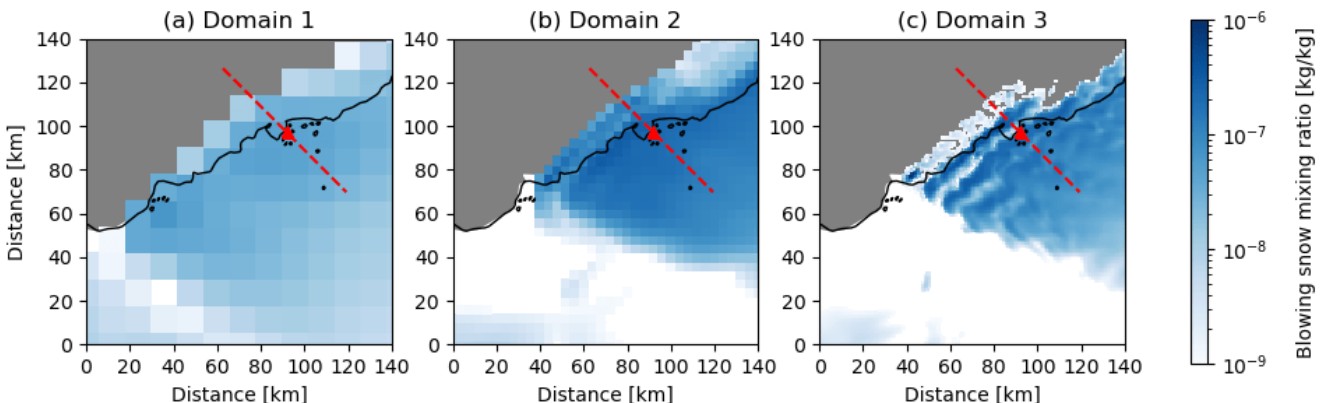

**Figure 11.** Blowing snow mixing ratio: Horizontal cross-sections at 1500 meters above ground level on the 10th of August 2017 at 1200 UTC for the three domains with (a) 12 km, (b) 4 km and (c) 1 km resolution. The red triangle indicates the location of the DDU station. The dashed red line indicates the location of the vertical cross-section shown in Fig. 12. Note that the colorbar is logarithmic.

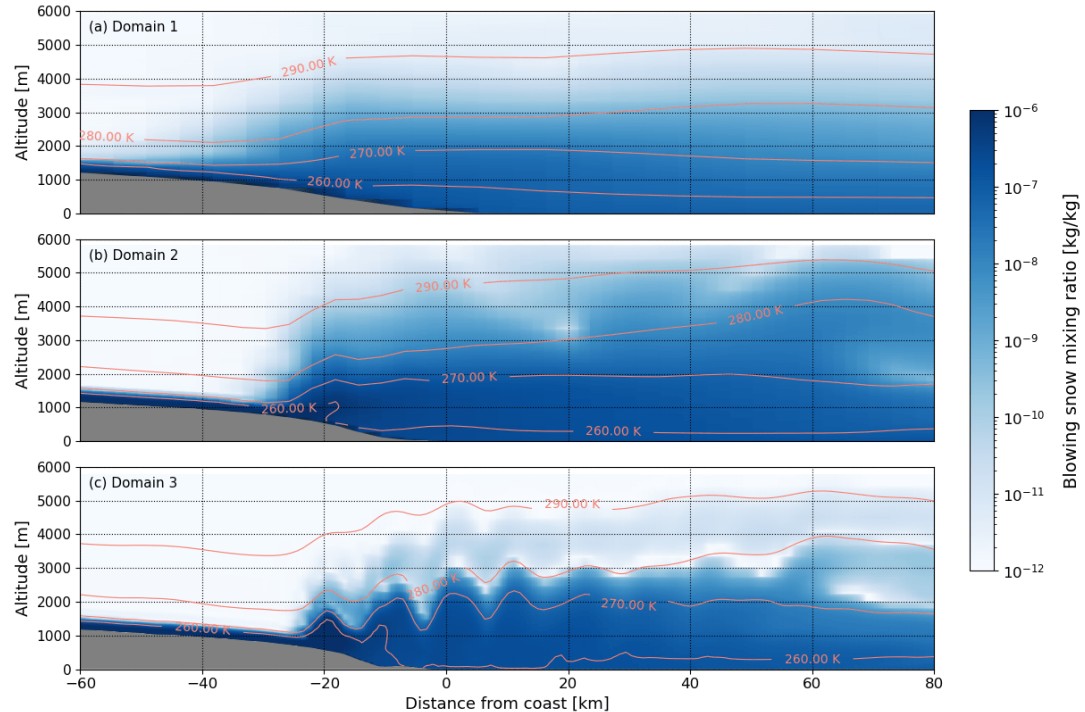

**Figure 12.** Blowing snow mixing ratio: Vertical cross-sections at the transect through DDU indicated by the dashed red line in Fig. 11 on the 10th of August 2017 at 1200 UTC for the three domains with (a) 12 km, (b) 4 km and (c) 1 km resolution. Four temperature contours are indicated (orange). Note that the colorbar is logarithmic.



### 4.3 Case Study II: Valley-scale simulation of a snow storm in the Swiss Alps

#### 4.3.1 Objectives

This case study showcases the ability of CRYOWRF to simulate alpine snowpacks in complex topography. It highlights the ability of CRYOWRF (on account of its WRF-based atmospheric core) to perform extremely high resolution, LES scale snow-atmosphere coupled simulations in addition to mesoscale flows using the same modelling framework. Due to the differences in spatial and temporal scales relevant for alpine meteorology as compared to polar meteorology, different snow
physics parametrizations, with higher precision are required for simulating alpine snowpacks compared to polar snowpacks. For example, this case study uses a different albedo parametrization than the previous, polar-based simulations. Melt water dynamics, new snow density parametrization, and snow-soil interaction are additional examples of processes that are more crucial in alpine environments as opposed to polar regions. This case study thus provides an additional template for interested future users of CRYOWRF in terms of namelist settings for SNOWPACK and WRF, more suitable for alpine environments.

In terms of novelty, the most interesting meteorological feature of this case study is the simulation of blowing snow in the Alps. While blowing snow in the Alps received significant attention in the cryospheric community from the surface mass balance (Vionnet et al., 2014; Gerber et al., 2018; Voegeli et al., 2016) and avalanche forecasting (Lehning and Fierz, 2008) perspective, its impact on different atmospheric components such as cloud dynamics or local radiative balance are unknown. There is evidence of potential feedback loops between blowing snow and cloud formation (Geerts et al., 2015; Vali et al.,
2012). However, to the best of our knowledge, it has not been explored numerically.

#### 4.3.2 Simulation Setup

The simulation consists of two domains with the larger domain at a spatial resolution of 1.25 km, covering an area of 225 square kilometers and the inner nest with a spatial resolution of 250 m, covering an area of 60 square kilometers. The outer domain covers a significant portion of eastern Switzerland and adjoining parts of Austria and Italy with the *alpine arc* passing
roughly across the middle of the outer domain. The alpine town of Davos is located in the south-west quadrant of the inner domain, offset by less that 5 km from the domain center. The location of the domains and the complex topography over which the simulation is conducted are shown in Fig. 13.

The simulations performed are similar to those by Gerber et al. (2018) in that they are driven by hourly analyses fields of the operational COSMO model (COSMO refers to COSMO-1 in this study). Underlying topography and landuse are from the Aster
1 s digital elevation model (Spacesystems and Team, 2019) and the Coordination of Information on the Environment (Corine) dataset (European Environmental Agency, 2006), respectively, prepared as WRF input as in Gerber and Sharma (2018). The difference is the use of CRYOWRF in this case study as opposed to standard WRF by Gerber et al. (2018). The boundary scheme in both domains is the Shin-Hong scale-aware scheme (Shin and Hong, 2015), which is applicable for resolutions as high as 250 meters. This is a unique scheme in the WRF physics suite that is able to represent subgrid-scale mixing in the *gray*
*zone* that exists between mesoscale flow $\mathcal{O}(1km)$ and turbulent flow $\mathcal{O}(10-100m)$. RRTMG is used for parametrization of





radiative effects both for short and longwave radiation, while the Thompson microphysics scheme is used to account for clouds. At the relatively high resolutions used in this case-study, no convection scheme is required.

As mentioned earlier, using SNOWPACK as the LSM requires per-pixel *.sno* files for initializing the snowpacks at each pixel. In this case study, snow profiles are generated using snow height and snow density information from the single-layer

data available in COSMO analysis fields and using 30 snow layers. The temperature of the layers is calculated via linear interpolation between the surface temperature and the underlying soil temperature, again using data from the COSMO fields. The initial grain diameter is arbitrarily chosen to be 300 microns with bond diameters of 75 microns for all layers. The main difference between this case study and those presented earlier is that the underlying soil column is accounted for as well with an additional 10 soil layers for each pixel. This is possible using SNOWPACK's existing capability for solving for both the soil

and snow columns and is crucial for using CRYOWRF for simulating seasonal snow covers and over regions with partial snow coverage.

### 4.3.3   Results

The above simulation was performed for a period of 36 hours beginning on the 11th of March, 2019 at 0000 hrs UTC for the inner, high resolution domain. This period is coincident with a period with high wind speeds recorded on alpine ridges

surrounding Davos. Furthermore, the wind direction is from the north-west, blowing across the inner simulation domain such that many of the ridges simulated in the region are nearly perpendicular to the dominant, large-scale flow direction. After allowing for an extensive period for spinup of the model, results are analyzed only for the period between 1800 UTC on the 11th of March, 2019 and 1200 UTC on the 12th of March, 2019.

To set the meteorological context of the case study, wind speed and temperature simulated by CRYOWRF at two locations,

the upper, Weissfluhjoch station (denoted by WFJ, 46.833323 N/9.806371 E, altitude: 2691 m ASL) and at Davos (denoted DAV, 46.802128 N/9.833477 E, altitude: 1594 m ASL) in the valley (is shown in Fig. 14). Two variants of the simulation, with and without blowing snow sublimation were performed to assess the impact of enhanced heat and mass transfer between the particles and the atmosphere. The wind speed within the valley is reasonably stable and moderate with values of approximately 7.5 m/s. On the other hand, at high altitudes, the wind is significantly faster at 1800 hrs UTC reaching speeds as high as 20 m/s

at WFJ. The wind speed at WFJ decelerates from this high value but remains significantly higher than those in the valley until 0400 UTC on the 12th of March. As expected, the air temperature is lower at WFJ than at DAV initially and the higher altitude regions experience significant warming on the 12th of March, 2019

Fig. 15 and Fig. 16 present surface maps of various diagnostic quantities relevant for blowing snow dynamics calculated in the innermost, 250 m resolution domain. In Fig. 15a patterns of cumulative erosion and deposition during the course of

the simulation are shown. It is instructive to compare the regions of extensive blowing snow erosion and deposition with the topography in Fig. 13c. The most active areas for snow transport are the ridges perpendicular to the flow as expected. It can be observed further that erosion occurs mostly on the wind-ward side of the ridges with zones of deposition found immediately on the leeward side, in agreement with earlier model assessments and measurements in the area (Mott and Lehning, 2010; Mott et al., 2010). On the other hand, there is not much blowing snow activity in the valleys. Note that qualitatively, there seems



to be more erosion than deposition. This could be explained by Fig. 15b, which presents vertically integrated mass loss (or gain) of blowing snow due to sublimation (or deposition). There are regions with extensive sublimation of blowing snow that are coincident with regions of high snow erosion, which is in partial contradiction to earlier model results (Zwaaftink et al., 2011) and needs further investigation. Interestingly, there is a significant zone of sublimation found along the windward edge of the Engadin, valley system east of Davos. Furthermore, the vapor mass transfer between blowing snow and the atmosphere is

spatially quite heterogeneous with vapor deposition being the dominant mass transfer in many regions, particularly in the center of the domain. The map of vapor mass transfer quantities and sign seems to suggest that the currently established paradigm of blowing snow largely undergoing sublimation may be too simplistic and may be valid only in terms of spatial and temporal averages with significant underlying heterogeneity in both rates and directions (transport of water from particles to air and vice-versa).

Fig. 16 illustrates the horizontal variations of surface friction velocity (subplot a) and blowing snow mixing ratio in the saltation layer (subplot b) at 0000 UTC on the 12th of March 2019. As expected in regions with complex topography, there is significant heterogeneity in surface shear stress. Saltation mixing ratio is a diagnostic quantity that is linked to the surface friction velocity and the state of the cohesion of the snowpack's surface through complex, non-linear relationships (see Sec. 2.3). Recall that the saltation mixing ratio acts as the boundary condition for blowing snow equations for snow in suspension. High

values of saltation mixing ratios are found in regions of topographic prominence such as ridges and the high altitude region in the center of the domain. However, there is no saltation occurrence in the lower lying parts of the domain. This is not only due to lower wind speeds but also due to the fact that the cohesion is higher at low altitudes due to warmer air temperatures.

    Saltation occurs very close to the surface, within a height of centimeters. However, snow in suspension, though many orders of magnitude lower in concentration undergoes long range transport, both horizontally and vertically. In classical boundary

layer meteorology, transport in the horizontal and the vertical is driven mostly by advection and turbulent mixing respectively. However, in alpine regions, the interplay between complex topography and atmospheric stability results in a wide range of complex flow types such as orographic waves, rotors, katabatic and anabatic winds etc. Such motions can be significant sources of vertical advection of blowing snow. These features are also active in preferential deposition of precipitating snow (Lehning et al., 2008; Gerber et al., 2018). Enhanced vertical displacement of blowing snow particles further increases their propensity

for horizontal transport due to higher wind speeds aloft.

    An illustration of the complexity of blowing snow clouds is shown in Fig. 17 which shows the blowing snow mass and number mixing ratios (subplots a and b, respectively) along a vertical cross-section located at the dashed red line shown in Fig. 13c. While blowing snow quantities are not significant from a surface mass balance perspective, they are relevant from a *microphysics* perspective. From this point of view, blowing snow particles can be found even 1000 meters above the surface,

at some locations with values of the order of 0.01 kg/kg. While it is not well known at the moment whether blowing snow particles trigger any microphysics feedback processes, for example acting as seeders for cloud formation, CRYOWRF can be used for further research in this direction. The final subplot in Fig. 17 shows the mean particle radius, which is a diagnostic variable from the blowing snow module. There is qualitatively a trend of decreasing radii with height even though there are significant variations from this. There are cases for example where even 100 micron particles are found at significant altitudes



above the surface. A video that shows the dynamics of the blowing snow mixing ratios for the entire simulation period is
provided as Supplementary Video M2.

There is an important caveat to values of blowing snow presented in this case study. As described in the previous section,
the snowpack was initialized using single-layer snow information from COSMO-1 analysis fields. Surface grain diameters and
cohesion between the grains were arbitrarily chosen. For very short simulations as in this case study, the initial conditions of

the snowpack can play a critical role in modulating the simulation results. The ideal way of initializing the snowpack would be
by first running ALPINE3D (the spatially distributed version of SNOWPACK) forced by meteorological or even COSMO-1
analyses fields for the domain of interest and initializing the snowpacks within CRYOWRF from the outputs of ALPINE3D.
This would provide a reasonably correct state of snowpacks at the start of the simulation. Such a workflow would be followed
in dedicated studies focused on verification of CRYOWRF for alpine simulations.





**Figure 13.** Location of the nested simulation domains for Case Study II. (a) Extent of the WRF domains with the location of Davos (DAV) shown (black triangle). (b) and (c) Underlying topography of the two domains respectively. The red dashed line across the domain in (c) is the location of the along-wind cross-sectional view in Fig. 17 and Supplementary video S2.



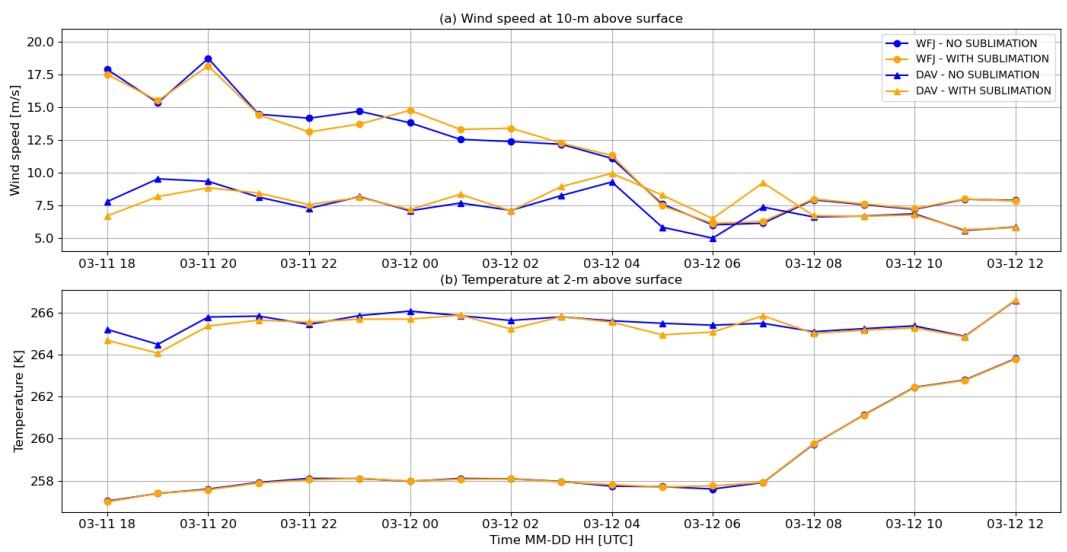

**Figure 14.** Time series of wind speed and air temperature as simulated by CRYOWRF at Weissfluhjoch (WFJ) and Davos (DAV) between 11th March 2019 at 1800 UTC and 12th of March 2019 at 1200 UTC.

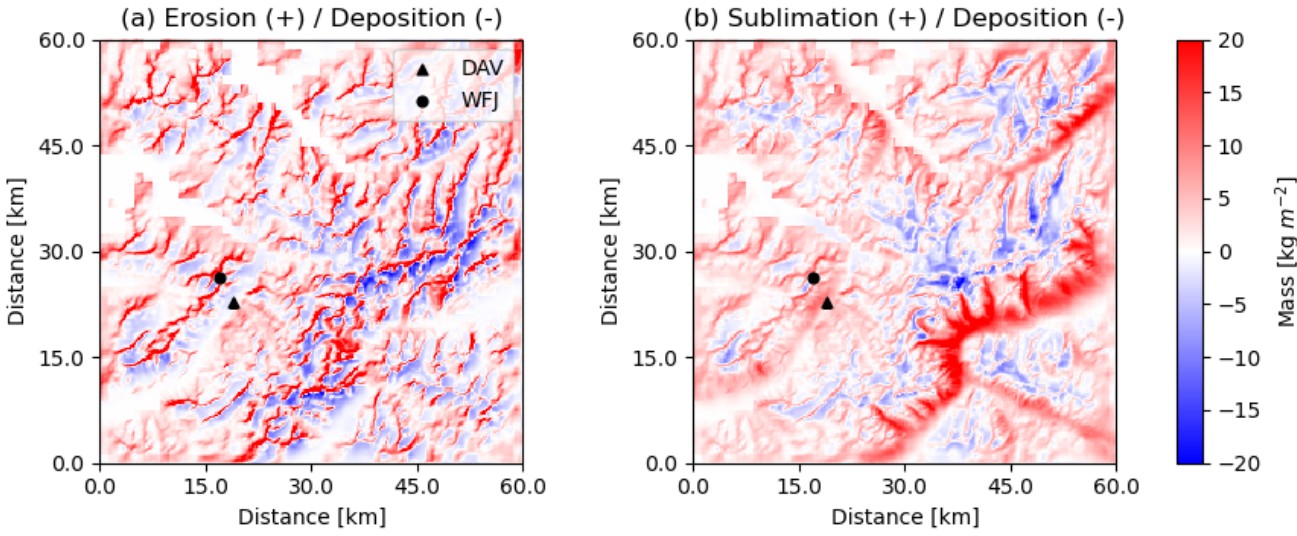

**Figure 15.** Impact of blowing snow: (a) Erosion and deposition of snow at the surface due to snow transport. (b) Sublimation (or deposition) of blowing snow particles in the atmosphere. Stations DAV and WFJ are marked by a black triangle and circle, respectively.





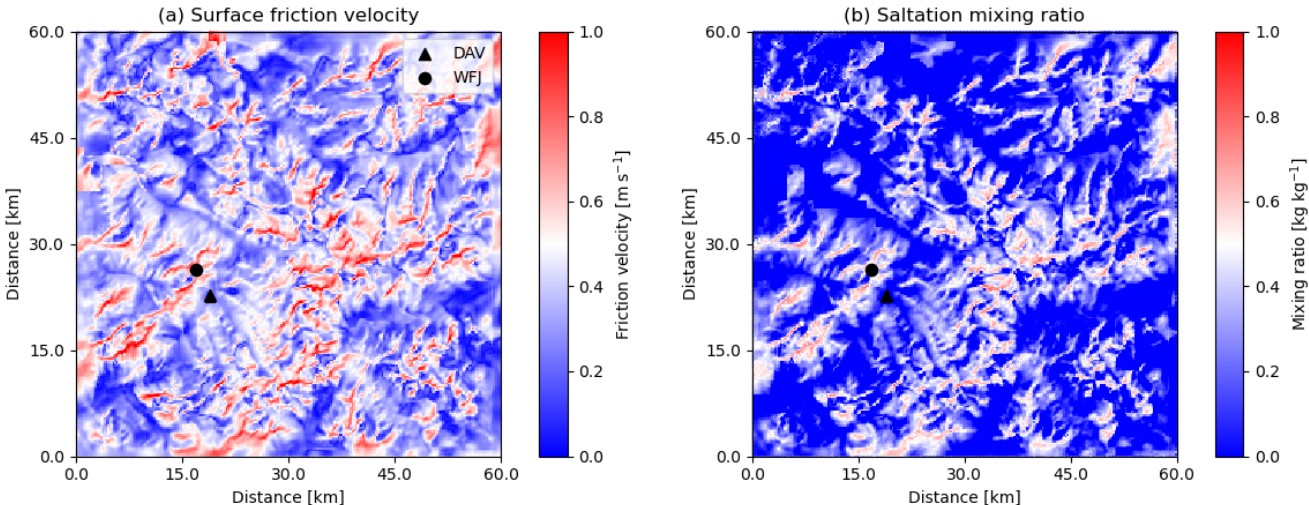

**Figure 16.** Modelling of snow transport: (a) Surface friction velocity that triggers saltation. (b) mixing ratio of blowing snow in the saltation layer. Stations DAV and WFJ are marked by a black triangle and circle, respectively. Results shown are at 0000 UTC on the 12th of March, 2019.



**Figure 17.** Blowing snow in the atmosphere on the 12th of March, 2019 at 0000 UTC along the transect marked by the dashed red line in

Fig: 13c. (a) Blowing snow mixing ratio, (b) number mixing ratio and (c) particle radii.



## 5  Summary and Outlook


This article describes the first version of CRYOWRF, a new modelling framework to simulate atmospheric flows and snow-pack dynamics in cryospheric environments. CRYOWRF is an extension of the widely-used atmospheric model, WRF, with an advanced complexity snow model, SNOWPACK acting as the land-surface model. Additionally, CRYOWRF includes a novel blowing snow scheme, a modelling capability not available in publicly released versions of WRF. The development of CRY-

OWRF was guided by three main design goals. First, that the entire workflow, from compilation, pre-processing, running the simulation to finally post-processing of simulation output must be as close as possible to standard WRF. The second goal was to ensure that all the technical capabilities of WRF such as hybrid (MPI+OpenMP) parallelism, multiple nesting, input/output and start-stop-restart capabilities were maintained. The final goal was to create a coupling framework such that CRYOWRF remains agnostic to the versions of the underlying WRF and SNOWPACK models.

Apart from a model description, three case studies were presented that showcase the suitability of CRYOWRF for simulating coupled atmospheric flow and snowpack dynamics over polar ice sheets on one hand (case study I) and snow-covered alpine topography on the other (case study II). The cases show that processes which have been inaccessible to previous mesoscale modelling efforts can be simulated using CRYOWRF. In case study Ia (Sec. 4.1), results from a year long simulation over the entire continent of Antarctica were compared with station measurements to establish the accuracy of the model. In case study

Ib, CRYOWRF was applied to simulate a complex atmospheric flow phenomenon involving the formation of a hydraulic jump and gravity wave excitation along the entrainment and vertical advection of blowing snow particles off the East Antarctica coast. This phenomenon can only be captured using resolutions of $\mathcal{O}(1\,\mathrm{km})$. None of the pre-eminent models for Antarctica research (RACMO/MAR) can simulate such phenomena due to their hydrostatic dynamical core - a limitation CRYOWRF does not have. The final case study (case study II, Sec. 4.3) showcases the applicability of CRYOWRF for simulations in snow

covered alpine terrain at resolutions of $\mathcal{O}(100\,\mathrm{m})$. Even though such simulations have been performed in the past with alternate modelling frameworks such as Meso-NH, standard WRF cannot perform such simulations due to the lack of a blowing snow model - a shortcoming that CRYOWRF remedies.

Apart from highlighting model capabilities, these case studies also serve as templates for potential users, who may wish to adopt CRYOWRF for their own research. All the necessary namelists, Python helper scripts to create the snowpack initial

conditions, post-processing scripts for data analysis and plotting, along with descriptions of the workflow for each of the these case studies is provided to act as *user guides*. The code is of course made publicly available (see data availability section for more details).

As this is the first version of CRYOWRF, there are many directions to take as far as the future road-map is concerned. One direction is to establish model accuracy through verification studies comparing model outputs with weather station and remote

sensing datasets. The analyses presented in case study Ia are being expanded with intercomparison with far more extensive measurement datasets and will be presented in a forthcoming publication. Case studies Ib and II are also starting points for more detailed analyses and can shed light on hitherto under-explored aspects of impact of blowing snow on components on the atmospheric system such as cloud dynamics and radiative effects. These setups need further verification.



The other direction, in a longer timeframe, is to expand the applicability of CRYOWRF to other areas of the cryospheric envi-
ronment not considered in this article. The two most promising areas for expansion is to test CRYOWRF for modelling sea-ice
atmosphere interaction as well as snow-forest-atmosphere interaction. In each of these two cases, the underlying SNOWPACK
model has been used in single-column, measurement-forced setups in previous publications (see the Introduction for refer-
ences), providing guidance in setting up CRYOWRF for simulations in these cryospheric environments.

The development of CRYOWRF can be viewed from various different perspectives. On one hand, it can be seen purely as
an improvement in the capabilities of the standard WRF for cryospheric regions. If existing users of WRF who are working
on topics related to cryospheric environments find CRYOWRF useful, that in itself would make the effort required for its
development worthwhile. However, the other, more general perspective is to consider CRYOWRF as model that combines a
mature, advanced atmospheric model core with a mature, high-complexity snow cover model allowing exploration of poorly
understood or novel cryospheric phenomena and feedback loops. From this perspective, CRYOWRF carries with it the promise
that the sum may be greater than its parts.

*Code availability.* CRYOWRF v1.0 is made available for public use using a GPL v3 License. The code is available at Sharma (2021b). The
institutional repository, consisting only of stable releases is located at https://gitlabext.wsl.ch/atmospheric-models/CRYOWRF. For obtaining
the code with most recent updates, download from https://github.com/vsharma-next/CRYOWRF

*Data availability.* In the interest of establishing reproducibility and for new users to adapt CRYOWRF for their own research, a dedicated,
citable repository for reproducing results in this article has been created here. See Sharma (2021a) for accessing the complete repository.

The reproducibility repository consists of two sections.

– Reproducibility (simulation): a collection of all namelists used in the case studies. It includes python helper scripts that create the ascii
  *.sno files containing initial snow profiles. These files are used to initialize snowpack at each pixel in the domain in CRYOWRF

– Reproducibility (postprocessing): a collection of data outputted from CRYOWRF simulations and python scripts to create all the figures
published in this article. The intention, apart from reproducibility is also to provide new users with scripts to post-process the new kind
  of datasets produced by CRYOWRF. For example, the snow profiles by CRYOWRF are not found in standard WRF and the provided
  script helps in understanding the encoded geometry of snow layers.

*Video supplement.* Two supplementary videos are included as a part of this article. Supplementary Video M1 shows the formation and
dynamics of *snow walls* formed at the DDU station in Anatarctica (see case study Ib, Section 4.2). Supplementary Video M2 shows the
blowing snow mass and number mixing ratio fields at a vertical cross-section across an alpine transect marked in red in Fig. 13c from case
study II



## Appendix A:  New namelist settings for CRYOWRF

CRYOWRF is a superset of WRF in the sense that it *adds* SNOWPACK as LSM to WRF. Recall that WRF is the *master* model and thus all WRF-SNOWPACK interactions are governed by WRF and its namelist. In Table A1, new namelist variables for CRYOWRF are listed. All the new namelist variables are in the *physics* section of the WRF namelist. Examples of namelists can be found in the *Reproducibility (Simulation)* dataset.

| namelist variable | value | purpose |
|---|---|---|
| sf_surface_physics | 18 | switches on SNOWPACK as LSM |
| blowing_snow | .true./.false. | switches on/off the blowing snow module |
| blowing_snow_sublimation | .true./.false. | switches on/off blowing snow sublimation |
| bs_rad_effect | .true./.false. | switches on/off, effect of blowing snow particles on radiative transfer (note that this is not tested for CRYOWRF v1.0) |
| sn_start_from_file | .true./.false. | Whether the snowpacks at each pixel are initialized using *.sno files or from either nearest-neighbour interpolation ( for nests for example ) or from WRF restart files ( when restarting simulations) |
| num_bs_sfc_layers | integer > 0 | number of layers in the fine mesh. See Fig. 1 for reference |
| no_snpack_lay_to_sav | integer > 0 | number of layers allocated in WRF to store and print snow-profiles |
| snpack_dt | integer > 0 | timestep of snowpack in seconds |
| snpack_write_dt | integer > 0 | time interval between transfer of snow profile data from SNOWPACK C++ objects to WRF's fortran arrays. Note that this process is expensive and thus the transfers are limited to the frequency of WRF's IO for snow profiles. |
| snpack_mode | 'land'/'antarctica' | to determine whether the snowpacks in the domain are seasonal or permanent snowpacks respectively. |

**Table A1.** New namelist variables for CRYOWRF

*Author contributions.*  VS conceptualized, designed and implemented CRYOWRF. FG prepared detailed post processing scripts for analyzing CRYOWRF data apart from being the primary *user/tester* of CRYOWRF. ML guided the project, particularly with respect to the SNOWPACK part of the development. All three authors contributed in writing the manuscript.

*Competing interests.*  No competing interests are present





*Acknowledgements.* We thank the **Swiss National Supercomputer Center (CSCS)** for generously providing computing resources through **projects s873 and s938**. Main funding was provided through the Swiss National Science Foundation **From cloud to ground: Snow deposition in extreme environments, grant 179130**. This project was partially funded by the ENAC big call grant of EPFL (project LOSUMEA). We thank our project partners in **LOSUMEA, Prof. Alexis Berne** and **Dr. Etienne Vignon** for many fruitful discussions and particularly for introducing the fascinating case of the airborne *snow walls* of Antarctica to us, which were simulated in case study Ib. We are indebted to **Dr. Peter Munneke** of the Institute of Marine and Atmosphere research (IMAU), Utrecht University for providing us with the IMAU-FDM data to initialize the continent-scale Antarctica simulation in case study Ia. CRYOWRF's SNOWPACK model is a modified version of the standard SNOWPACK as developed by **Dr. Christian Steger** (now at ETHZ) and **Dr. Nander Weaver** (now at University of Colorado). We thank them for sharing their modified code with us. We thank our team members at CRYOS, EPFL, particularly **Mahdi Jafari** and **Adrien Michel** who are active SNOWPACK developers and users and to **Daniela Brito Melo** for proof-reading early versions of this manuscript. Finally, we are immensely grateful to the Free and Open-source software community. We made extensive use of the **wrf-python** and the **xarray** packages and all figures were prepared using **matplotlib**. Atmospheric weather station (AWS) data for the stations Concordia and Eneide were obtained from 'MeteoClimatological Observatory at MZS and Victoria Land' of PNRA - http://www.climantartide.it". Data for the AWS station D-17 was kindly provided by **Dr. Charles Amory**. For the station South Pole we use measurement data provided by the NOAA Global Monitoring Laboratory. Furthermore, the authors appreciate the support of the Automatic Weather Station Program for the AWS data sets of Mizuho, D-10, D-47 and D-85 (**Prof. Matthew Lazzara**, NSF grants number ANT-0944018 and ANT-1924730). For AWS data of the Kohnen station we kindly thank the IMAU and in particular **Dr. C.J.P.P. Smeets** for preparing and providing the data. For AWS data for the stations Casey, Davis and Mawson we thank the Australian Antarctic Data Centre, the Australian Antarctic Division and the Bureau of Meteorology, Australia. We kindly thank the AWI for providing AWS data of the Neumayer station on the PANGAEA data publisher for Earth & Environmental Science platform. Finally, we would like to acknowledge KU Leuven, and **Prof. Nicole van Lipzig** and **Dr. Alexandra Gossart** in particular, for providing us with AWS data of Princess Elisabeth station.





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
