# Peer review of "Introducing CRYOWRF v1.0: Multiscale atmospheric flow simulations with advanced snow cover modelling"

_Geoscientific Model Development, 2021_

## Author Comment (AC1)

**RESPONSE to Comments by Referees**

**Opening Remarks:**

We thank both the reviewers for their remarks and comments. The overall remarks support the publication of the manuscript in GMD and we are thankful for their positive remarks.

Both reviewers have a common refrain that is with regards to model validation. We understand the need for much more validation and we express as such repeatedly in the manuscript. However, the purpose of this article was to describe a new model implementation, our motivations for developing it, its design goals and examples to highlight their potential use. There are indeed a lot more validation studies in the works including one that should be submitted for peer review soon. However, such additions would take us further away from the scope and spirit on this article. We hope the reviewers find our view on this regard acceptable.

We additionally thank the reviewers for their detailed reading of the manuscript and raising some pertinent issues. Below, we present a point-by-point response (in light green) to the comments including the changes that may have been introduced in the manuscript in response to each comment (in red).

**RESPONSE To Comments by Anonymous Referee #1**

**Paragraph 1**

Sharma and colleagues present a novel coupled modelling system called CRYOWRF that consists of three components: the atmospheric model WRF, the snow model SNOWPACK, and a new parameterization for snow drift. Only a handful of previous studies have attempted to improve the representation of cryospheric processes in WRF by integrating new modelling components, and there has been no publicly available implementation of blowing snow to date. The authors present three case study simulations across a wide range of horizontal grid spacings, with the associated namelists and scripts provided as templates to facilitate usage of the model by the scientific community. As such, the manuscript fits the scope of Geoscientific Model Development well, and provides a significant advancement in the field of coupled atmospherecryosphere modelling. Overall, the paper is very well written and organized. The methods are generally well explained, although I have highlighted a few aspects that would benefit from additional clarification in the minor comments. The main weakness of the paper is the dearth of model evaluation. There is also an issue in relying on asynchronous coupling (i.e., not calling SNOWPACK every WRF timestep) for computational efficiency if one is interested in investigating feedbacks, as mentioned in the major comments below.

**Our Response:**

We thank the reviewer for the positive feedback and for supporting the publication of the manuscript in GMD.

**Major comments**

1. The introduction does not mention previous efforts to improve the representation of cryospheric processes in WRF through integration of new modelling components, including Collier et al. (2013; https://doi.org/10.5194/tc-7-779-2013) and Eidhammer et al. 2021 (https://doi.org/10.5194/hess-25-4275-2021).

Thank you for pointing us to these references. It must be noted however that Eidhammer et al. is not a 'coupled' model. Their implementation of CROCUS is in WRF-Hydro and as far as we understand their work, their experiments would be considered as 'offline' modelling. Thus, we have included the Collier et al. reference which in fact is quite illuminating.

2. In order to reduce the computational overhead of integrating SNOWPACK, the authors suggest to use, and present case studies that employ, asynchronous coupling between WRF and SNOWPACK through the namelist parameter snpack\_dt. From the cryospheric perspective, there is no clearly no need to call the snow model every timestep (i.e., every 5 s in a 1-km grid spacing domain). However, from the atmospheric perspective, the difference in the update frequency of turbulent heat fluxes and surface conditions will introduce numerical artefacts that are unrelated to the improved representation of cryospheric processes. The reliance on asynchronous coupling therefore limits the utility of CRYOWRF as a tool to investigate feedbacks, in particular between "offline" simulations with other LSMs and "online" simulations with SNOWPACK. This limitation should be clarified in the manuscript.

We agree with your comment to the point that difference in the update frequency of turbulent heat fluxes and surface conditions *may* introduce numerical artefacts. However, we would like to point out that the asynchronous time-stepping is a *capability* that is available to the user. It is the user's choice whether to employ it or not. One may choose snpack\_dt to the same as WRF's timestep. The code would work seamlessly. The number of layers in SNOWPACK can be set to an upper limit to reduce the computational cost in such as case.

An additional point is that deciding whether to call SNOWPACK every *WRF* timestep and the number of layers to use in SNOWPACK depends on the aim of the numerical experiments. Experiments to study snow-atmosphere interaction where turbulence is resolved (in LES mode) are orthogonal to seasonal simulations on continental scale. However, we clarify and re-emphasize this point in the manuscript as follows:

**(In Section 2.2)**

A crucial technical capability of the SNOWPACK model is the possibility for *smart* merging (splitting) of vertical layers based on their similarity (strong gradients). The most important use of this capability is to allow the user to set a maximum number of layers or maximum snow depth to simulate.

**(In Section 3)**

It is important to note that the capability of using different timesteps for SNOWPACK and WRF is a choice provided to the user. This capability must be juxtaposed with the capability of SNOWPACK to simulate a constrained number of layers or snow height. Calling SNOWPACK at larger intervals would result in degrading the quality of surface fluxes while keeping the computational cost low. However the larger timestep would only be required if the goal of the study is to study the snowpack at very high vertical resolution thereby necessitating hundreds of layers of snow. As described earlier, the goal of CRYOWRF is to be useful for simulations at vastly different spatial and temporal scales. The motivations for simulations at different scales are typically orthogonal. For example, one may choose to use CRYOWRF in a turbulence resolving large-eddy simulation (WRF-LES mode) to study snow-atmosphere interaction. In such a numerical experiment, it may be prudent to call SNOWPACK every timestep while simulating only a couple of snow layers. On the other hand, a user perhaps wants to perform continental scale simulations for surface mass balance where it is imperative to use a large number of snow layers where calling SNOWPACK every timestep is not necessary. Future studies using CRYOWRF in different experiments would ultimately result in coming up with a set of best practices for each such case.

3. With the exception of Figures 5 & 6, there is no model evaluation presented, and this task is repeatedly designated as future work. Figures 5 & 6 compare simulated near-surface meteorological variables with station data for the first case study (an analysis that is later stated as "establishing the accuracy of the model" at line 709), however there is no evaluation of surface mass balance, or more importantly, snow drift as simulated by the new parameterization. The manuscript would be strengthened by additional evaluation, even if suitable data are only available on a point scale (e.g., for blowing snow).

In the process of addressing this point (and even in earlier drafts of the manuscript), we added more intercomparisons with station measurements including datasets of blowing snow measurements. Ultimately, we reached the conclusion that it would be more suitable to keep the current publication as a technical article with detailed model description along with sample simulations as showcases and keep intercomparisons for a separate publication – this publication

is soon going to be submitted for peer-review. Incidentally, this reasoning is provided in the manuscript already in the opening paragraph of Section 4.

4. In several places in the manuscript, the authors credit CRYOWRF with capabilities that are actually provided by WRF regardless of LSM choice (e.g., lines 411—412; line 552; lines 713-714; WRF is acknowledged at line 587). Therefore, the manuscript would benefit from more careful language around the value added by CRYOWRF.

Our principal thrust for CRYOWRF development to enhance WRF to match the capabilities of the pre-eminent models in Antarctic Research (RACMO and MAR). In the process, we realized that due to WRF's dynamical core, we had infact surpassed the limitations of the above models, opening new avenues for research (the hydraulic jump with entrained blowing snow being one such example). We further realized that alpine meteorology and surface-atmosphere interaction in the alpine regions could benefit from CRYOWRF's blowing snow model. We state exactly these thoughts in the manuscript (at lines 712 – 717) as:

"None of the pre-eminent models for Antarctica research (RACMO/MAR) can simulate such phenomena due to their hydrostatic dynamical core - a limitation CRYOWRF does not have. The final case study (case study II, Sec. 4.3) showcases the applicability of CRYOWRF for simulations in snow covered alpine terrain at resolutions of O (100 m). Even though such simulations have been performed in the past with alternate modelling frameworks such as Meso-NH, standard WRF cannot perform such simulations due to the lack of a blowing snow model - a shortcoming that CRYOWRF remedies. "

We doubt that any reader would *credit* CRYOWRF with any capabilities that the reviewer feels we have unjustly claimed as being our unique development. It was certainly not our intention to do so and in fact would be quite foolish to attempt such a thing. WRF is perhaps the most well-known weather model with well established capabilities. We have stated repeatedly, beginning with the

- abstract ("CRYOWRF couples the state-of-the-art and widely used atmospheric model WRF, with the detailed snow-cover model SNOWPACK. "),

- in the description of WRF model in Section 2.1 ("Additional important technical capabilities are ease of performing multi-scale atmospheric simulations with both one-way and two-ways nesting, and start-stop-restart capabilities for all nests. The multiscale capabilities and its community driven nature made WRF the ideal choice as the atmospheric core of CRYOWRF.")

- and even more explicitly in Section 3 on model capabilities and performance ("Significant effort was also expended to ensure that CRYOWRF adopts all the capabilities of the parent WRF and SNOWPACK models ").

But coming more directly to the point: CRYOWRF does indeed have all the capabilities as claimed. This was a choice and not a given. For example, it would

have been easier to implement the snowpack coupling, the I/O and restart and to a minor extent, the blowing snow model for single domain simulations and not for multiple nested domains (which technically enable WRF's multiscale, highresolution simulations). Whether WRF is the 'provider' for claimed capabilities is not the important point here, but the fact that CRYOWRF sustains and not diminishes those capabilities.

5. It would be helpful to clarify already in the methods section that SNOWPACK can function as a standalone LSM, and therefore also updates surface conditions and fluxes over non-glacierized grid cells. For readers unfamiliar with SNOWPACK, this capability is unclear until lines 623-626.

We thank the reviewer for pointing this out. We agree with the suggestion and have modified section 2.2 as follows.

"As SNOWPACK accounts for soil and vegetation layers, it can be used as a fullfledged, standalone land surface model (LSM) even for non-glaciated terrain and indeed has been used as such in the form of its spatially distributed version, Alpine3D"

Minor comments

1. Please provide references for the statements at lines 16-17 & lines 592-593.

One reference has been added for lines 16-17. In our opinion, there are no references necessary for lines 592-593. Those lines are our own, opiniated statements.

2. Lines 316-317: Please provide more information about how the stability correction is handled to avoid runaway cooling in the interactive implementation.

We are unable to respond to this comment because we did not quite grasp what is meant by it – "runaway cooling" ? "Interactive implementation"?

3. Line 385: Is it correct that only those three variables – latent & sensible heat fluxes and surface albedo – are updated in WRF? If so, why aren't other surface boundary conditions updated, like surface temperature and roughness?

Other surface 'boundary' conditions like surface temperature and roughness are updated as well. However, for the principal prognostic equations, only sensible and latent heat fluxes are directly required. Note that for glaciated pixels, a fixed roughness of 0.002 m is used. The updated manuscript reads as: " The forcings from WRF are the incoming shortwave and longwave radiation along with temperature, relative humidity and wind speed and SNOWPACK returns to WRF, the sensible and latent fluxes along with albedo and surface temperature. Note that in the current model, the surface roughness length is kept fixed for glaciated pixels. Thus, the update for surface roughness length occurs only upon transition from non-glaciated to glaciated (typically upon snowfall) and vice-versa (typically upon melt) "

4. Section 4.1.1: Why was this simulation period selected?

This review was selected because we wanted to initialize our simulations with RACMO's output. There were two outputs offered to us. Either July 2010 or July 2016. We decided to use the July 2010 output as it would give us an opportunity to not only provide an example for this manuscript but also a good starting point for a decadal simulation for a future article (the same article with much more extensive validation as described earlier). Already pre-empting a possible follow-up question: why RACMO's outputs? The RACMO project has expended significant effort in coming up initial conditions using their IMAU-FDM model in standalone model. For our initial studies, we felt it would be prudent to use their outputs as a starting point for our studies.

5. Section 4.1.4: Could the authors discuss why SNOWPACK improves on the warm bias simulated by the Noah-MP LSM?

No. Unfortunately we could not find the reason for this improvement apart from only speculative guesses such as the fact that CRYOWRF includes the effect of blowing snow sublimation which our simulations with NoahMP do not. Secondly, answering this question would require a deeper understanding of exactly what NoahMP is doing, the albedo model used therein, a comparison of exchange coefficients etc etc. Doing such an analysis would take us away from the purpose of the article. However, we add the following lines in the manuscript as:

"The improvement of CRYOWRF with respect to NoahMP could be due to a multitude of reasons. One reasoning for example, could be that CRYOWRF includes the effect of blowing snow sublimation which has a cooling effect on the atmosphere. However, this is mere speculation and a deeper analysis would be necessary to explain these differences, which is out of scope for this study."

6. Line 529: Could the authors provide a justification for only calling SNOWPACK every 15 minutes? From the cryospheric perspective, this timestep is nearly 10x larger than a characteristic timescale for heat diffusion assuming a top layer height of 1 cm. From the atmospheric perspective, this is a 180x decrease in the frequency of updating surface fluxes and conditions in the 1-km domain (dt=5s).

The intent of the simulation is to capture a purely mechanical phenomena over a period of 3 days where the intention is not to study near surface meteorology or surface and sub-surface thermodynamics. It is a study where only the 'fast' dynamics are important. Thus, calling the LSM every 15 minutes is sufficient.

As a generic response apart from this specific case study, while the reviewer is correct in comparing the characteristic timescales of thermal diffusion to the timestep, it can be argued that the thermal diffusion is an order term in determining the thermal budget of the top layer. We have used 15 minutes in most cases as that timestep is typically sufficient in offline studies for snowpack simulations.

In response, we have added the following lines:

"Using a lower timestep for the LSM is justified since the phenomenon we are interested in simulating is a purely mechanical one and not particularly dependent of the near-surface heat fluxes, thermal snow-atmosphere interaction and certainly not the surface and sub-surface thermodynamics."

7. Line 687: Another important caveat would be that there has been no evaluation of the results.

Added the following:

"There is an important caveat to values of blowing snow presented in this case study (apart from the lack of verification with field measurements)"

**Technical comments**

1. Line 46: Please rephrase "not to speak about"

The phrase is reframed as -

"This is due to neglecting the strong vertical gradients of density and temperature typically found in snowpacks and spatially heterogeneous phase changes."

2. Line 129: "OpenMP"

Corrected.

3. Line 351: OOP has not been defined

Corrected and modified as:

" Secondly, as SNOWPACK follows the 'object oriented programming' (OOP) paradigm, the coupling library allocates and maintains pointers to SNOWPACK objects during runtime."

4. Line 391: Please rephrase "performed using Noah-MP along with CRYOWRF" to clarify that separate simulations were performed and compared.

Corrected and modified as:

"A baseline simulation was performed using Noah-MP and compared with separate CRYOWRF simulations with 10, 50, 100 and 400 layers of snow respectively."

5. Line 479: "Sublimation"

Corrected.

6. Line 490: "period is between"

Corrected.

7. Line 495: DDU has not been defined in the text

Corrected and title modified as:

"Case Study Ib: Multiscale simulation at Dumont d'Urville: formation of hydraulic jumps"

8. Line 500: "topographic" Corrected.

**Paragraph 1**

Blowing snow and the associated sublimation for snow redistribution is an important process to incorporate in polar atmospheric models especially those that can capture fine spatial scales through consideration of nonhydrostatic dynamics, as done here with the WRF model. This represents an advance on existing capabilities with the hydrostatic models RACMO and MAR. One can only appreciate the impacts of blowing snow and the associated sublimation by doing runs with and without the blowing snow processes active, not done here.

This is an interesting manuscript on coupled atmosphere-snow cover modeling and its impacts that deserves to be published after some improvements. The "land surface" model implemented into WRF is SNOWPACK. The blowing snow scheme implemented is similar to Dery and Yau (2002) with differing treatments for terminal fall velocities of snow particles and thresholds for onset of snow transport from the surface.

**Specific Comments:**

1. No mention is made of Polar WRF that has pioneered the use of WRF in the polar regions, adapted and added physics treatments, and provided guidance on the parameterization performance in high latitudes, underlying the Vignon et al. (2020) manuscript that is featured prominently here. A place to start is here: http://polarmet.osu.edu/PWRF/

We have added a reference to Polar WRF in the manuscript. However, we would also like to note that our development incidentally began with Polar WRF until we realized that most of Polar WRF's 'improvements' were geared towards sea ice and ice surfaces. One important addition of Polar WRF, the ability to ingest sea-ice fields as time-dependent surface boundary conditions from external forcing data in fact became a part of 'standard' WRF. Apart from sea-ice and ice surface related modifications in Polar WRF, we are unaware of any other significant difference between Polar WRF and standard WRF. Our contention is that Vignon et al. (2020) could have been performed with standard WRF without any difference ( we are confident of our contention as Etienne Vignon's work was conducted as a part of our collaborative project LOSUMEA as described in our acknowledgements). We have added the following lines as :

"We also acknowledge the pioneering work in polar meterological modelling using the polar optimized version of WRF named Polar-WRF \citep{hines2008}.

Polar-WRF has adapted existing physical models in WRF, undergone extensive evaluations and added technical capabilities (such as ingesting frequent sea surface temperature and sea-ice masks), many of which are now a part of \textit{standard} WRF. The most important contribution of Polar-WRF and associated studies is to compile \textit{best practices} for polar modelling using mesoscale models that have inspired various other models including our development. "

2. The big differences between CROWRF and WRF with NoahMP at South Pole (Fig. 6) are the large warm biases of the latter during the warmer part of the year and the much higher relative humidity values during winter. Any explanations? These biases are much larger than previous implementations of WRF/Polar WRF over the Antarctic: doi: 10.1029/2012JD018139. Moisture content of the air is challenging to measure at the low air temperatures at South Pole in winter. Are you certain that the higher relative humidity values simulated by WRF with NoahMP are not more correct? It is often thought that the air there is close to or exceeds saturation with respect to ice. Are your relative humidity values with respect to ice or liquid water? Surface pressure, 10-m wind speed, and 10-m wind diection are much closer, and consistent with previous Antarctic WRF studies.

Yes, we too were a bit surprised by the poor performance of NoahMP at the South Pole. However, unless there is some tweaking on NoahMP required that we are unaware of, these are the results as obtained from the simulations. One important point to note is that the plot is for potential temperature and not air temperature. Since the altitude of the South Pole is significant, any differences between air temperature would be inflated when computing the differences between potential temperatures.

We have added the data for observations for relative humidity. Now if the observations themselves are erroneous must be investigated. However, these values are quality controlled. If the observations themselves are to be questioned (inspite of the fact that it is indeed very hard to get good measurements of humidity in cold temperatures), it is hard to say which models are 'better'. That said, most models (apart from NoahMP) do predict over saturation during the winter which is not borne by the relatively sparse observational datasets of this quantity. Two additional factors complicating an intuitive analysis is that 2-m quantities are typically diagnostics and due to complicated near-surface phenomena like blowing snow thermodynamics, the picture is especially more complicated in Antarctica.

The relative humidity values are with respect to ice. Thank you for pointing out that this was missing in the manuscript and is now added.

3. The surface mass balance components shown in Fig. 7 look in error to me. If the mean values listed are averages for all of Antarctica including the ice shelves then precipitation and surface mass balance are only 2/3rds the values given by van Wessem et al. (2018) for long-term averages that should approximate the values here. Does the sublimation refer to total values, i.e., blowing snow plus surface sublimation? Do you really think that large melting and refreezing is widespread over the two large ice shelves (up to 200 kg/m\*m/yr)? These are cold even in summer. Melting does occur in summer but is limited on average. https://doi.org/10.1002/2013GL058138

There is significant interannual variability for mass balance quantities in Antarctica. We are not surprised that the values are different from van Wessem et al (2018)'s long term averages. For example, 2012 was one of the snowiest years on East Antarctica (specially in Queen Maud Land) with precipitation much more that the cumulative sum of the several preceding or subsequent years. Hence the increasing research in atmospheric rivers – a significant source of interannual variability in precipitation. The only true measure of van Wessem's or any other (surface mass balance) SMB studies is comparison with either GRACE data or stake measurements. In a following article we perform such validations. However, to include them in this article would be beyond the scope of the article or the purpose of GMD.

With regards to melting and refreezing, there are many regions of Antarctica where melting may exceed 100 kg/m2/yr (or 100 mm per year S.W.E). In fact such values are found even in the paper the reviewer has added (at Larsen for example). The second important point is the net balance between melt and refreeze is of importance. From the numerical modelling perspective, melting, which typically happens at the surface is quite sensitive to the top layer thickness. The top layers in our example are quite thin (less than a centimeter) whereas in RACMO, MAR, the top layer is typically 5 cm thick. In spite of this numerical feature, the thin layers also result in immediate refreezing. The net result of melt and refreeze (not showed in the manuscript ) should typically be comparable between models and qualitatively, this would be the case but has not been evaluated here.

4. Incorporatethismanuscriptintoyourpaper:https://doi.org/10.1029/2020JD033936

Added

5. Line 142: Add "atmospheric" before "stability corrections".

Corrected

6. Line 433: Provide details about the vertical levels used in the model: How many? Vertical distribution? What is the lowest level? What is the highest level?

Added as:

7. Line 479: "Sublimation".

Corrected

8. Line 549: "as well as an acceleration". Don't understand what is being said here.

We meant acceleration of wind speeds. Stands corrected in the updated manuscript.

9. Line 576: "replace "detained" by "detailed".

Corrected.

10. Fig. 12 caption: Make clear that the contours are potential temperature. Corrected. Thank you for catching this.

---

## Author Response (AR2)

SECOND REVISION

RESPONSE to Comments by Anonymous Referee #2

Below, we make an attempt to respond to the comments and critiques of the referee point-by-point

1. Specifically this relates to Figures 5 and 6. According to this new manuscript https://doi.org/10.1007/s11707-022-0971-8, WRF-NoahMP has a large warm bias over Antarctica due to its deficient treatment of surface albedo, being too low and therefore absorbing too much shortwave radiation in summer and making the near-surface temperatures too warm. This is almost certainly the cause of its deficient temperature simulation, especially for South Pole. The authors don't state why they use potential temperature instead of physical temperature, but the latter is to be preferred. If physical temperatures were plotted then one could see whether the relative humidity values produced by WRF-NoahMP happened at low air temperatures. I have seen humidity sensors become insensitive at temperatures approaching -35C, and basically reflect the air temperature, i.e., it is conceivable that WRF-NoahMP is correct in its RH output, rather than the observations.

The referee seems to be making two points here: (a) Noah MP in WRF has been shown to have a warm bias over Antarctica based on a recent paper and (b), NoahMP being correct in its RH output rather than the observations at low temperatures.

The goal of this paper is to present a new modelling framework in WRF with an advanced snow model and blowing snow. We successfully did the technical work and presented a snapshot of results supporting this. We use NoahMP as a baseline setup, the observations as reference and data from our new model and compare the results 'out-of-the-box'. We show not just for South Pole but also 14 other stations (spread over the continent) in Figure 4 that the new modelling framework produces results as good as NoahMP if not better 'out-of-the-box'. It is not the goal of the paper, or indeed our research to try and understand why NoahMP or any other competing LSM model produces the results that they do. Neither is the goal of the article to understand and solve the long-standing problem of humidity measurements at very low temperatures.

We created a new coupling framework, publicly released it and wrote an article showcasing its capabilities in three very different scenarios for processes occurring at very different length and timescales. The reviewer would agree that if we enter into discussions on the details, this article could conceivably be split into three separate manuscripts! And we indeed will follow up with many

further manuscripts to address the finer details. We don't feel appropriate to enter into these discussions in this article.

However, to satisfy the reviewer, we have added a reference to the article they link to and mention clearly the problem with measurements at low temperatures.

2. I continue to be concerned about the surface mass balance results shown in Figure 7 and whether the big discrepancy with long-term averages can be ascribed to interannual variability. At the very least discussion of this issue should be included in the manuscript rather than just ignoring it.

We have never ignored it! We stated clearly in the submitted manuscript:
"CRYOWRF outputs these quantities as two-dimensional fields directly as a part of the standard output via its specially implemented diagnostics package. An example of the surface mass balance output during this case-study is shown in Fig.\ref{fig:fig_7}. Since the simulation period is of only one year, it is not possible to quantitatively compare the results presented in this figure with those published in literature, which are typically calculated over climate time scales of at least 30 years."

We have added a line stating clearly how the long-term averages can be different from the those from this one year.

"… which are typically calculated over climate time scales of at least 30 years. In fact, given the interannual variability of SMB patterns, particularly at the coast (due to phenomena such as atmospheric rivers), it is likely that the numbers from one year as shown in Fig.\ref{fig:fig_7} may be different from long-term averages."

To assuage the concerns that the reviewer has with regards to the accuracy of CRYOWRF, we add below a figure from a recently submitted article (the pre-print should be available soon publicly) where the results of CRYOWRF are compared to MAR for a simulation period of a decade between 2010-2020. The reviewer can see that CRYOWRF's numbers are not only comparable to MAR but in many ways improve upon MAR in comparison to stake measurement-based SMB (which in our opinion is the most valid source of SMB data as opposed to other models including RACMO / MAR etc).

[Figure]

*Figure 1: All rights belong to Gerber, Sharma and Lehning: CRYOWRF - a validation and the effect of blowing snow on the Antarctic SMB (2022), (submitted to JGR Atmospheres / ESSOAR )*

3. Finally, I didn't find any manuscript response to my point number 6 about model vertical levels.

    Yes, this is indeed a mistake from our side. We have now added this information to each of the three case studies. We apologize for overlooking this question in the previous round of reviews.